# How Far Ahead Do LLMs Plan?
# Uncovering the Latent Horizon in Chain-of-Thought Reasoning

**Liyan Xu** [1]  **Mo Yu** [1]  **Fandong Meng** [1]  **Jie Zhou** [1]

## Abstract

Chain-of-thought (CoT) reasoning has become a central mechanism for eliciting multi-step reasoning in Large Language Models (LLMs). Yet recent evidence presents a tension: hidden states appear to already encode future reasoning before CoT fully unfolds, while explicit steps still remain crucial for tasks requiring compositional computation. To deepen the understanding between LLM's internal states and its verbalized reasoning trajectories, we investigate the latent planning strength of LLMs, through our probing method, Tele-Lens, applying to hidden states across diverse task domains. Our empirical results indicate that LLMs exhibit a *myopic* horizon, primarily conducting incremental transitions without precise global planning. Leveraging this characteristic, we propose a hypothesis on enhancing uncertainty estimation of CoT, which we validate that a sparse set of *pivot* positions can effectively represent the uncertainty of the entire path. We further underscore the significance of exploiting CoT dynamics, and demonstrate that automatic recognition of CoT bypass can be achieved without performance degradation. Our code, data and models are released at https://github.com/lxucs/tele-lens.

## 1. Introduction

Chain-of-Thought (CoT) (Nye et al., 2021; Wei et al., 2022) has fundamentally reshaped problem-solving in natural language processing, marking a shift from traditional pattern-matching approaches, e.g. encoder-based classification (Devlin et al., 2019; Liu et al., 2019), toward prompt-based reasoning articulated explicitly in natural language (Zhou et al., 2023; Dong et al., 2024; Sahoo et al., 2025). The

[1] WeChat AI, Tencent Inc. Correspondence to: Liyan Xu <liyanlxu@tencent.com>, Jie Zhou <withtomzhou@tencent.com>.

*Proceedings of the 43rd International Conference on Machine Learning*, Seoul, South Korea. PMLR 306, 2026. Copyright 2026 by the author(s).

capacity of CoT is further amplified through extensive thinking emanated from reinforcement learning, characterized by recent models such as DeepSeek-R1 (DeepSeek-AI, 2025).

While CoT is widely perceived as the de facto reasoning paradigm, however, recent studies on Large Language Models (LLMs) have revealed complementary, and at times seemingly conflicting perspectives. On the one hand, LLMs have been shown to exhibit **internal planning on the reasoning trace prior to the explicit emergence of CoT**. (Pal et al., 2023) reveals that early hidden states already encode information about subsequent generations; (Sheng et al., 2025) finds that reasoning capability is largely pre-planned before generation begins; (Dong et al., 2025) observes that early hidden states can reliably predict reasoning steps and key attributes with high correlation. Similarly, numerous studies have shown that the initial stages of CoT can effectively anticipate the final answers (Azaria & Mitchell, 2023; Gottesman & Geva, 2024; Afzal et al., 2025).

The internal planning of LLMs appear to diminish the significance of CoT generation, raising the question of whether the thinking process is just echoing pre-determined paths already encoded in prior internal states. On the other hand, theoretical analyses advocate that **CoT is indispensable due to the limited expressivity of Transformers** bounded by its architectures (Bhattamishra et al., 2023; Merrill & Sabharwal, 2023; Li et al., 2024), and only intermediate steps of CoT can derive length generalization (Anil et al., 2022; Xiao & Liu, 2025) and compositional reasoning (Wies et al., 2023; Abbe et al., 2024; Zubic et al., 2025). Therefore, the manifestation of pre-calculated trajectories appear *unlikely* via internal planning before the onset of CoT.

Nonetheless, the relationship between the model's internal representations and its verbalized reasoning tokens largely remains opaque. In this work, we investigate the ***internal dynamics of CoT***, and target the following questions concerning the latent planning horizon:

Q1. *To what extent do hidden states encode a global plan for the reasoning roadmap, as opposed to supporting rather local, incremental state transitions?*

Q2. *How does the planning horizon further influence other characteristics of CoT reasoning?*

We believe answering these questions is important not only for deepening our understanding of CoT dynamics, but also for providing critical insights into broader model-thinking design scenarios. For example, recent works leverage LLM internal states to enable early exiting from CoT reasoning, thereby mitigating the *overthinking* problem (Yong et al., 2025; Yang et al., 2026). Recently, both GPT-5[1] and Claude Code[2] introduce forms of *adaptive thinking* that route user requests to models of different sizes or allocate varying levels of reasoning effort. Such mechanisms fundamentally rely on how strong the model can "sense" or "see through" the input complexity through internal planning.

Towards this objective, we derive empirical insights by examining the synergy between explicit CoT steps and its latent planning horizon. Building on the observations, we then highlight the significance of leveraging CoT dynamics on estimating CoT's uncertainty and necessity.

To answer the first question, Section 2 presents a series of probing experiments designed for dissecting LLM hidden states, aiming to evaluate the internal planning strengths with respect to future reasoning trajectories. We first introduce our probing method, termed **Tele-Lens**, which employs a trained low-rank adapter (Houlsby et al., 2019) that transforms each hidden state within CoT steps to predict Teleological information along multiple dimensions, including subsequent tokens, final answers, reasoning lengths, etc. Importantly, unlike prior works that primarily address single-domain tasks, we conduct probing experiments across 12 diverse datasets spanning different classes and domains, ranging from straightforward knowledge question answering to classic hard problems, e.g. Parity (counting the number of digits as even or odd), a canonical challenge for Transformers (Chiang & Cholak, 2022; Hahn & Rofin, 2024).

By empirical results, we observe sharply contrasting behaviors across probing dimensions and task domains, as detailed in Section 2.5. For instance, while early hidden states may encode predictive signals of the final answer for relatively simple tasks, they can behave nearly randomly on compositional tasks, where they only begin to reliably capture the precise answer within the final one or two reasoning steps before CoT completion.

Overall, our probing results bring a unified view of the prior complementary beliefs from previous works: LLMs exhibit a ***myopic planning horizon***, in which hidden states primarily support immediate, local transitions rather than long-range, global trajectories. However, for simpler tasks that fall within the LLM's single-step pattern-matching capacity, early hidden states may capture a coarse gist of the

---

[1] https://openai.com/index/gpt-5-system-card
[2] https://platform.claude.com/docs/en/build-with-claude/adaptive-thinking

final answer—albeit in a more heuristic manner, rather than through a precise, pre-planned reasoning process.

Addressing Q2, leveraging latent CoT signals, we first focus on **uncertainty calibration** over CoT, where an effective confidence metric, e.g. the rollout perplexity or entropy, should assign high scores to correct reasoning paths and low scores to uncertain ones (Huang et al., 2024; Chen et al., 2024; Bakman et al., 2025). We propose a hypothesis followed by empirical validation: the uncertainty of CoT follows a **Wooden Barrel principle**. Just as the capacity of a barrel is determined not by its average stave height but by its shortest stave, the reliability of a reasoning chain is governed by a small number of *pivot* positions. Intuitively, as the model's latent planning is *myopic*, most CoT tokens are high-confident local transitions that may dilute the underlying uncertainty of the entire trace. We therefore speculate that focusing on a small set of *pivot* positions instead of global aggregates could enable more precise uncertainty estimation. Empirical results in Section 3.1 find that even a simple strategy of top-$k$ selection can effectively enhance the accuracy of estimation across all three general uncertainty metrics, yielding up to 6% absolute improvement.

Beyond uncertainty estimation, we also present a *proof-of-concept* that the CoT planning patterns can be leveraged to recognize whether CoT is necessary to derive the final answer, achieving automatic **CoT bypass** that directly outputs the answer with minimal performance degradation. Experiments in Section 3.2 demonstrate that our proposed strategy using Qwen3-32B can realize up to 16.2% CoT bypass with only a negligible 0.03 overall accuracy drop.

Back to Q2, our proposed strategies on CoT uncertainty and necessity estimation further underscore the significance of analyzing CoT dynamics, which encode hidden yet valuable information. We hope that this work on uncovering the latent planning horizon could advance the understanding of CoT synergy, and spur more identification of hidden signals to be exploited in broader model-thinking scenarios.

## 2. CoT Planning Horizon

This section delineates the detailed experimental setup and findings on diving into the latent planning capacity. Section 2.1 introduces our probing method, Tele-Lens, followed by a description of the data setup and model configurations. Empirical results are reported in Section 2.5.

### 2.1. Tele-Lens

To enable probing across multiple dimensions, including subsequent token prediction, our method is designed to support prediction over the full LLM vocabulary. To this end, we adopt a transformation-based approach to probe various Teleological information upon the CoT trace, dubbed Tele-

Lens. It follows the paradigm of Logit Lens (nostalgebraist, 2021; Belrose et al., 2023), originally for examining layer-wise interpretability in Transformers, which bridges the hidden states from intermediate Transformers layers to the final LM head directly, thereby enabling whole-vocabulary prediction. For Tele-Lens, to mitigate overfitting and computational overhead, we adopt a bottleneck low-rank adapter (Houlsby et al., 2019) with added nonlinearity for hidden state transformation, more formally described as follows.

Concretely, for an LLM rollout, we denote the response tokens in its thinking process up to the final answer as $T = \{t_1, t_2, .., t_n\}$, representing a reasoning trajectory of length $n$ (throughout this paper, we use the terms "thinking" and "CoT" interchangeably). The hidden state corresponding to token $t_i$ at the $k$-th Transformers layer is then denoted as $H_i^k \in \mathbb{R}^d$, with $d$ being the LLM hidden size. The corresponding transformed hidden state $\widetilde{H}_i^k \in \mathbb{R}^d$ by applying the bottleneck adapter, and its predicted probability distribution $\mathcal{P}_i^k$ over the LLM vocabulary $\mathcal{V}$, are defined as:

$$\widetilde{H}_i^k = \mathrm{GeLU}\Big(\big(H_i^k + \mathrm{Emb}^k(\delta)\big) A^k\Big) B^k \quad (1)$$

$$\mathcal{P}_i^k(\mathcal{V} \mid t_i, A^k, B^k, \mathrm{Emb}^k, \delta) = \mathrm{Softmax}\big(\widetilde{H}_i^k L\big) \quad (2)$$

where $A^k \in \mathbb{R}^{d \times r}$, $B^k \in \mathbb{R}^{r \times d}$ and $\mathrm{Emb}^k \in \mathbb{R}^{m \times d}$ are the learnable parameters of the adapter for the $k$-th Transformers layer, typically with a low rank $r < d$. Particularly, $\mathrm{Emb}^k$ is an *optional* embedding matrix, taking an offset $\delta = 1, 2, .., m$ to inject the target predicting position up to $m$ offset. $L \in \mathbb{R}^{d \times |\mathcal{V}|}$ is the LM head matrix that will keep frozen during adapter training.

For each token $t_i$ in the reasoning path, we take its hidden state of each Transformers layer and probe along three teleological dimensions:

- **Subsequent tokens**: we use solely $H_i^k$ to predict its $m$ following tokens $\{t_{i+\delta} \mid \delta = 1, 2, .., m\}$. Each offset $\delta$ is injected respectively as in Equation (1).
- **Reasoning length**: we use $H_i^k$ to predict the total length of the thinking. Instead of applying LM head by Equation (2), we take $\widetilde{H}_i^k$ followed by a single regression layer to yield a number prediction.
- **Final answer**: we use $H_i^k$ to predict the final answer directly, with $\mathrm{Emb}^k$ removed in Equation (1). Each answer should be uniquely identifiable by a token in $\mathcal{V}$, thus this suits only for tasks with a fixed answer space.

If LLMs maintain a precise global plan early in the reasoning process, then the strength of that plan should, in principle, be measurable by these probing targets, before explicit CoT fully unfolds. For final-answer probing particularly, as answer tokens constitute the natural continuation of the reasoning process, they can reflect how early, and to what extent, future outcome information is internally represented throughout CoT generation.

## 2.2. Tasks and Datasets

As previous works on CoT analysis mainly focus on specific domains of interest, the findings can be complementary that reflects different perspectives and angles, as discussed in Section 1. Towards more comprehensive empirical insights, we broaden the scope of domains and include 12 diverse tasks, which we categorize into three types as below. Concrete examples of these tasks are provided in Appendix A.1.

**Explicit Compositional Tasks**  These tasks require explicit multi-step procedures to resolve, involving a high degree of structural modularity. Notably, as suggested by both prior empirical studies and theoretical analyses, Transformers often struggles to efficiently perform function composition within a single forward pass (Dziri et al., 2023; Merrill & Sabharwal, 2023; Zubic et al., 2025). Consequently, such tasks usually require intermediate CoT steps to derive the final answer. We include the following three tasks, for which data generation is fully controllable.

- **Parity**: a classic task often seen in Transformers' expressivity and learnability analysis (Chiang & Cholak, 2022; Bhattamishra et al., 2023; Hahn & Rofin, 2024). Given a sequence of digits, the task essentially asks whether the total count of a target digit is even or odd.
- **Cycle**: we adopt a task introduced by Abbe et al. (2024), in which the input consists of a list of directed edges, forming either a single full-sized cycle or two half-sized cycles. The task requires determining whether there exists a path between two specified vertices, or equivalently, whether they fall into the same cycle.
- **Subsum**: an algorithmic task, Max Subsequence Sum, adopted in prior Transformers studies (Dziri et al., 2023). For a list of $n$ numbers, the task computes the maximum sum of its subsequences, which admits an $O(n)$ dynamic programming solution. We query the least significant digit of the maximum sum for a fixed answer space.

**Implicit Compositional Tasks**  These tasks typically require multiple reasoning steps as well, but in a more nuanced and implicit manner embedded in the problem semantics, such as mathematical or logical reasoning. For math-related tasks, we adopt three following datasets: **GSM8K** (Cobbe et al., 2021), **MATH** (Hendrycks et al., 2021b), **AIME** (AIME, 2025). To enable a fixed answer space tailored for final-answer probing, we adapt each problem into a multi-choice format, by prompting GPT-4.1 to generate plausible yet misleading distractor options. Details on the multi-choice conversion are provided in Appendix A.3.

For logical reasoning, we include the following two datasets that evaluate soft reasoning framed in natural language: **MuSR** (Sprague et al., 2024), **Zebra** (Lin et al., 2025).

**Knowledge and Semantic Tasks**  These tasks primarily focus on knowledge-intensive queries grounded in the pro-

vided semantic context, without a particular focus on intense reasoning. This category comprises four datasets: **CSQA** (CommonsenseQA, Talmor et al. (2019)), **MMLU** (Hendrycks et al., 2021a), **QuALITY** (Pang et al., 2022), **GPQA** (Rein et al., 2024). Brief descriptions of all existing datasets are further provided in Appendix A.2.

As all 12 tasks have a fixed answer space, with most of them being multi-choice questions, each answer is uniquely identifiable by a token in the vocabulary. Accordingly, the label set for the final-answer probing is constituted by 20 tokens in total, as detailed in Appendix A.3.

### 2.3. LLM Backbones

To obtain response rollouts and corresponding hidden states, we consider two types of LLM backbones described below.

**Off-the-Shelf LLM**    As our probing experiments require access to both model weights and reliable CoT outputs, we employ the open-source Qwen3 series with native support of both thinking and non-thinking modes. We use **Qwen3-32B** as the primary backbone to ensure robust performance while maintaining manageable computational cost.

**In-Domain LLM**    In addition to open-source LLMs with readily available thinking modes, we also employ an in-domain LLM trained with task-specific supervision, for two key reasons. First, a task-aware model exhibits more stable and decisive reasoning, thereby serving as an "upper bound" on internal planning capacity. Second, this setup helps reduce potential confounding factors inherent to general-purpose LLMs tied to specific model families.

Our In-Domain LLM learns task-aware CoT via reinforcement learning with GRPO (Shao et al., 2024). We intentionally train from **Qwen2.5-7B-Instruct**, which does not have thinking mode natively, allowing for a cleaner bootstrap of CoT behavior on these tasks. We introduce our detailed GRPO training settings in Appendix B.

### 2.4. Experimental Settings

**Dataset Construction**    We construct our probing datasets with train/dev/test splits across the 12 tasks, which contain up to 4000 / 100 / 500 problems per task, respectively. For the three tasks—Parity, Cycle, and Subsum—the problems are obtained via data generation. For other tasks, problems are sampled from their original datasets. Details of our data generation and sampling, as well as further statistics are provided in Appendix A.4 and A.5.

**Training and Hyperparameters**    For each probing dimension, we train a dedicated Tele-Lens adapter for each Transformers layer of a LLM backbone, using a rank of $r = 256$. Each training run is conducted for approximately 5K steps,

with early stopping on the dev set. More hyperparameters for adapter training are provided in Appendix A.6.

### 2.5. Empirical Results

We first report the performance of LLM backbones to characterize the 12 tasks, evaluating off-the-shelf Qwen3 and our trained In-Domain LLM. Full results are provided in Table 5 (Appendix B.2), from which we draw the observations:

- For those compositional tasks requiring explicit multi-step reasoning, direct answering without CoT can only achieves near-random performance (e.g. Parity, Cycle), corroborating prior findings on the expressivity limits of Transformers (Chiang & Cholak, 2022; Merrill & Sabharwal, 2023; Zubic et al., 2025). For other tasks, CoT generally yields substantial improvement as well.

- Owing to differences in model generation and scale, our in-domain LLM underperforms the naive Qwen3 models on certain datasets. Despite this, it achieves the best performance on three compositional tasks and attains overall performance comparable to Qwen3, while producing substantially shorter CoT trajectories (approximately 1K+ characters per CoT, compared to 10K+ for Qwen3). These results validate the training effectiveness in inducing **more stable and decisive reasoning paths**. A qualitative CoT comparison is provided in Appendix B.3. The latent planning horizon of In-Domain LLM is thus viewed as an "upper bound" for these tasks.

With Tele-Lens adapters trained and evaluated on the collected CoT trajectories, we present the empirical observations for each probing dimension as follows.

#### 2.5.1. PLANNING FOR FINAL ANSWERS

Figure 1 first presents the average probing accuracy with In-Domain LLM along the initial CoT positions (full results across all tasks in Figure 13). The overall trend with Off-the-Shelf Qwen3 is also similar as in Figure 14. At first glance, it is clear that different Transformers layers exhibit varying predictive capacities. Notably, the highest performance does not occur at the final layer, but rather at layers between the middle and the last, which is consistent with findings of prior works that intermediate layers encode richer semantic information (Reif et al., 2019; Garí Soler & Apidianaki, 2021; Skean et al., 2025). For analyses in this section, we focus on results by layer 48 (64 total) for Off-the-Shelf Qwen3 and layer 21 (28 total) for In-Domain LLM.

It is worth noting that although Figure 1 plots only the initial positions, the probing is performed over the complete CoT trajectories. We further present the accuracy evolution along the full individual reasoning traces in Figure 2, Figure 15 and Figure 16. Overall, we draw two key findings as below.

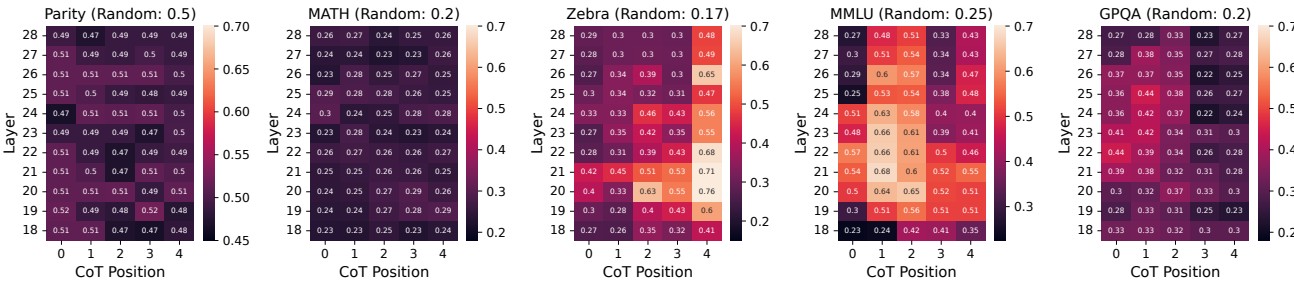

*Figure 1.* Results for the final-answer probing: average accuracy of In-Domain LLM for the first five tokens within CoT trajectories, measured across selected Transformers layers and tasks. The full figure across all tasks is presented in Figure 13 (see Appendix C).

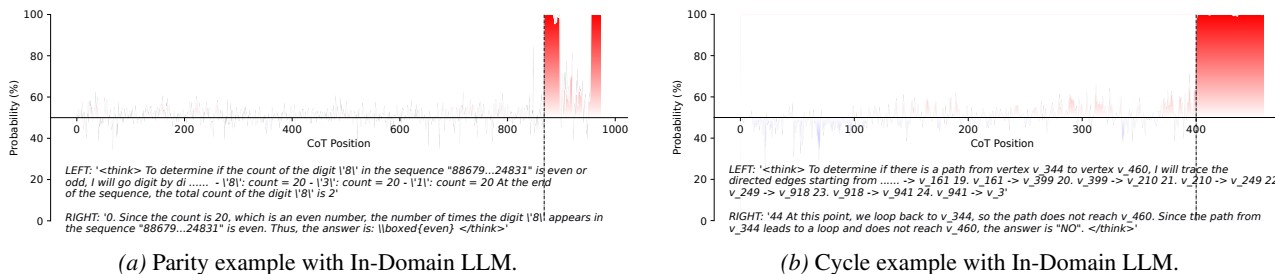

*(a)* Parity example with In-Domain LLM.        *(b)* Cycle example with In-Domain LLM.

*Figure 2.* Examples of final-answer probing accuracy along full CoT trajectories with In-Domain LLM (random guessing is at 50%). The vertical dashed line indicates the position at which accuracy first spikes. "*LEFT*" and "*RIGHT*" at the bottom illustrate the reasoning details right before and after the accuracy spike, respectively. Similar examples with Off-the-Shelf LLM are provided in Figure 15.

▶ *For precise future planning, LLMs exhibit a myopic horizon rather than long-term foresight.*

To illustrate this, we focus on explicit compositional tasks, where their initial final-answer planning is near random, shown by Parity, Cycle and Subsum in Figure 13&14. Analysis of the full planning dynamics, depicted by the two examples in Figure 2, reveals that the precise final-answer only emerges one step before the reasoning completion, such that the probability of final answers remain flat before the final spike in the end: the final answer for Parity is planned only after the counting of all digits, and for Cycle, it is planned only after observing a complete path or cycle. As answer tokens are natural continuations of the CoT reasoning process, they act as anchors that reflect short-sighted internal planning, which fails to capture multi-step foresight.

To demonstrate quantitatively, we parse the CoT trajectories and obtain the final-answer probability at each critical intermediate steps. For Parity, we report the probabilities of CoT positions right after counting each digit. As shown in Table 1, the probing only converges at the final counting position, exceeding 90%, while for preceding positions, the accuracy hovers around random guessing as 50%.

For simpler tasks, the probing also only stabilizes near the reasoning end. More illustrations are provided in Figure 16.

▶ *LLMs can exhibit coarse signals for final answers in early stages of CoT, but reflecting only a pattern-matching gist, rather than precise reasoning plans.*

*Table 1.* Probing probabilities for Parity at CoT positions immediately following the counting of each of the last four digits in the sequence (random as 0.5). Position **0** denotes the highest probing probability after all digits have been counted (the upper bound).

|  | -4 | -3 | -2 | -1 | **0** |
|---|---|---|---|---|---|
| In-Domain LLM | 0.49 | 0.51 | 0.51 | 0.97 | **0.99** |
| Off-the-Shelf LLM | 0.50 | 0.52 | 0.51 | 0.94 | **0.97** |

As shown in Figure 1, LLMs can sometimes "sense" the gist of the answer early on, particularly for those emphasizing semantic understanding rather than explicit multi-step reasoning. To illustrate with more clarity, Figure 3 depicts the probing dynamics on CSQA and MMLU, which focus more on semantics and knowledge, in which an early spike in probing accuracy is notably evident, in contrast to tasks like Parity. By the full evaluation results presented in Figure 17 and Figure 18, it appears that early hidden states do possess certain information predictive of the final answers, just as observed in prior works (Azaria & Mitchell, 2023; Gottesman & Geva, 2024; Afzal et al., 2025).

However, our in-depth analysis suggests that these coarse predictive signals primarily reflect a vague perceptual cue, but not resulting from exercising a pre-planned reasoning path. We proceed to compare the performance of this early coarse signal, with that of true reasoning via CoT, as well as direct answering without CoT; the results are presented in Figure 4 (full results in Figure 19). Across almost all tasks, early final-answer planning yields lower task accuracy

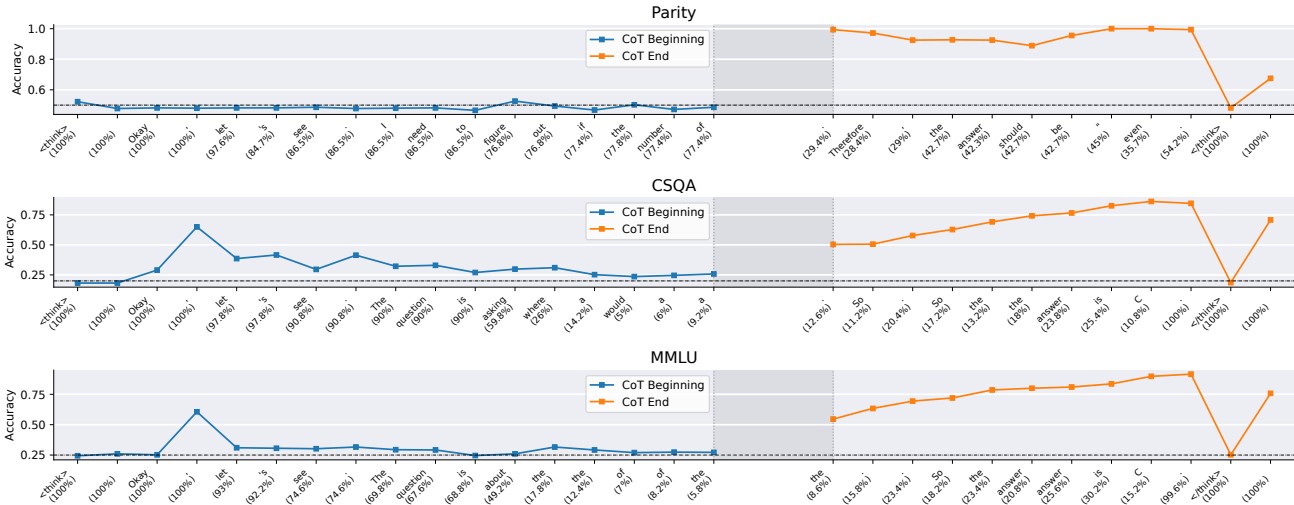

*Figure 3.* Average final-answer probing accuracy on three tasks with Off-the-Shelf LLM (Qwen3-32B) along CoT positions. The most frequent token at each position is annotated with its occurrence frequency. The notably earlier accuracy spikes are especially pronounced for Knowledge and Semantic tasks, but largely remain flat for Compositional tasks. The full results across all tasks are shown in Figure 17 for Off-the-Shelf LLM and Figure 18 for In-Domain LLM (Appendix C) .

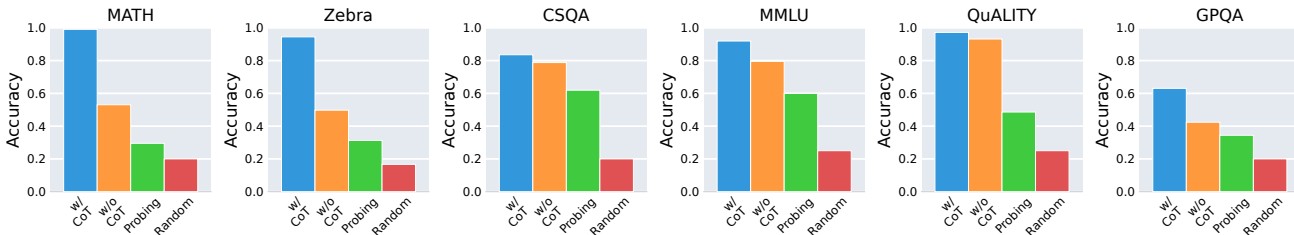

*Figure 4.* Task accuracy comparison for Off-the-Shelf LLM (Qwen3-32B) under four settings: using thinking mode (**w/ CoT**); using non-thinking mode (**w/o CoT**); the best probing accuracy among initial CoT positions (**Probing**); the random-guess baseline (**Random**). The coarse signals of early final-answer planning are shown inferior to the direct prediction counterpart without CoT involved. Full results across all tasks are provided in Figure 19. Similar comparisons for In-Domain LLM is provided in Figure 20.

than both standard reasoning with CoT and direct answering without CoT. Therefore, even with comparable reasoning budgets, early planning remains less effective than direct answering. The performance gap further widens substantially when CoT is applied, strongly indicating that such coarse signals **do not arise from precise plans in latent space**.

### 2.5.2. PLANNING FOR REASONING PATH

Empirical results on probing subsequent tokens further advocate a myopic planning horizon of LLMs detailed below.

▶ *LLM hidden states generally encode limited foresight over subsequent reasoning paths.*

For each hidden state, we assess subsequent token prediction performance up to its 8-th following token along the CoT trajectory. As LLM generation is a sampling process over a latent distribution, we measure by Top-5 Accuracy, deeming a prediction correct if the true subsequent token appears within the top-5 predictions. Figure 5 presents the evaluation results with In-Domain LLM, which show a clear

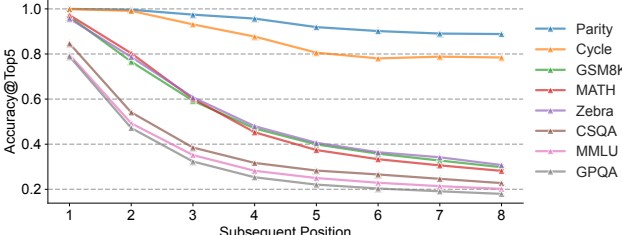

*Figure 5.* Top-5 accuracy for subsequent token prediction, using the last Transformers layer of In-Domain LLM. Full results across layers and tasks are presented in Figure 21 and Figure 22.

overall decline in accuracy as the subsequent token position advances, specially for tasks dominated by semantic understanding and factual knowledge (e.g., MMLU and GPQA).

Figure 5 also suggests that LLM does plan the subsequent path to a certain extent, with Top-5 accuracy exceeding 50% for the next two steps. However, a more long-term planning is only limited to tasks with structural modularity, such as Parity or Cycle, whose reasoning trajectories exhibit discernible patterns (see the CoT example in Figure 10). In



*Figure 6.* Heatmap of the predicted reasoning length (*y*-axis) using initial CoT hidden states against the actual reasoning length (*x*-axis). The rather unreliable predictions on those tasks without structural solutions (e.g. CSQA) suggest that a precise global plan does not emerge early in CoT, even for the task-aware In-Domain LLM. Full results across tasks are provided in Figure 23 and 24.

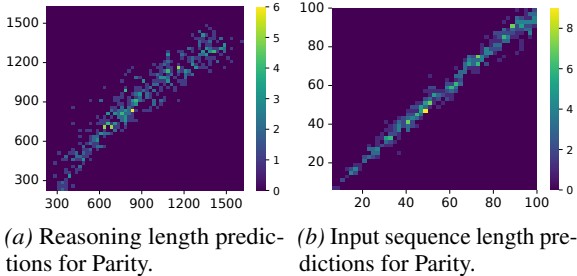

*(a)* Reasoning length predictions for Parity.   *(b)* Input sequence length predictions for Parity.

*Figure 7.* Task-specific factors can confound reasoning length predictions. For Parity, the total length is typically proportional to the input sequence length, which can be perceivable by LLMs.

general, hidden states lack a clear vision over subsequent reasoning. Beyond In-Domain LLM as an "upper bound", a similar trend is also observed for Off-the-Shelf LLM, albeit with much lower accuracy across all tasks, especially with significant drop on structural tasks, as illustrated by the comparison between Figure 21 and Figure 22.

### 2.5.3. PLANNING FOR GLOBAL STEPS

Reasoning length probing again indicates a lack of global planning prior to the emergence of CoT, as discussed below.

▶ ***LLMs do not grasp the reasoning length precisely, though task-specific heuristics may offer shortcuts.***

In general, if LLMs possessed a global reasoning view in sight, early hidden states would be predictive of the total length across input domains. However, our empirical results suggest that the initial CoT hidden states hardly have reliable internal clock for global reasoning length, for both In-Domain and Off-the-Shelf LLMs, as illustrated by the unstable and often low correlations across most tasks, shown by the heatmaps in Figure 6 (full plots in Figure 23 and 24).

On closer inspection, two tasks appear to be exceptions, Parity and Subsum, which exhibit high correlation with the true reasoning lengths, as in Figure 23. However, interpreting this as evidence of robust CoT planning on these tasks can be misleading. We highlight the attribution of task-specific confounding factors, illustrated in Figure 7: for both tasks, reasoning paths are typically in proportional to the input

sequence length, which is readily observable by LLMs and thus could serve as a shortcut signal in probing. In contrast, for Cycle as in Figure 6, such shortcut does not apply, as its reasoning length scales with the path between two vertices rather than the input length (example in Figure 11), which LLMs have difficulty estimating directly. The discrepancy between Parity/Subsum and Cycle further underscores the limited presence of actual global planning in LLMs.

## 3. Leveraging CoT Dynamics

Given the *myopic* planning horizon observed in our probing experiments, we highlight the significance of exploiting such CoT dynamics, and demonstrate how these planning characteristics can be leveraged to estimate both the uncertainty and the necessity of CoT.

### 3.1. CoT Uncertainty Estimation

For language models, general metrics such as perplexity or entropy are standard to estimate the inference confidence. A well-calibrated uncertainty metric should ideally assign high scores to correct outputs and lower scores to uncertain ones. In our studies, we target metrics that utilize internal signals within CoT trajectories, focusing on the three general uncertainty metrics described below.

(i) *Perplexity*, equivalent to the average Negative Log-Likelihood of the sequence (NLL) for uncertainty. For a sequence $X$ with $N$ tokens $\{x_1, x_2, .., x_N\}$, NLL is:

$$\text{NLL}(X) = -\frac{1}{N} \sum_{i=1}^{N} \log P(x_i \mid x_{<i}) \qquad (3)$$

(ii) Average *entropy* H across tokens in the sequence, with $\mathcal{V}$ being the model vocabulary:

$$\text{H}(X) = \frac{1}{N} \sum_{i=1}^{N} \left( - \sum_{w \in \mathcal{V}} P(w|x_{<i}) \log P(w|x_{<i}) \right) \quad (4)$$

(iii) *Self-Certainty* (SC) (Kang et al., 2025), defined on the

vocabulary distribution as below:

$$\text{SC}(X) = \frac{-1}{N|\mathcal{V}|} \sum_{i=1}^{N} \sum_{w \in \mathcal{V}} \log\left(|\mathcal{V}| \cdot P\left(w \mid x_{<i}\right)\right) \quad (5)$$

Note that the three metrics all aggregate over entire CoT positions in the sequence. Intuitively, however, tokens that steer the reasoning process within CoT are often sparse; the majority of tokens function as "syntactic fillers" necessary for linguistic coherence. These filler tokens are usually high-confidence local transitions, as evidenced by the density distributions presented in Figure 12, aligned with prior findings that most tokens in LLM are of low entropy (Wang et al., 2025b; Li et al., 2026b).

**Wooden Barrel Principle**    Building on the local planning cues in Section 2, we speculate that the internal signals localized around a few critical tokens are more informative than trajectory-wide aggregates, where a conventional global averaging across all generated tokens may **dilute the sensitivity** of the uncertainty estimation. We thus posit such principle: just like a barrel's capacity is determined by its shortest stave, analogously, we hypothesize that the uncertainty of a reasoning chain is governed a subset of critical logical leaps, which we term *reasoning pivots*.

We conduct empirical validation and demonstrate that focusing on these pivot positions, even through a simple top-$k$ selection strategy, could yield cleaner signals for uncertainty calibration. Our validation utilizes two orthogonal sources of latent signals, as described below.

**Latent Signals by Tele-Lens**    Before we proceed with general metrics of uncertainty estimation, we first demonstrate that latent signals from a sparse subset of tokens are indeed effective to characterize the uncertainty of the whole reasoning trajectory. Motivated by Figure 2, where specific positions exhibit significant accuracy spikes during final-answer probing, we propose utilizing the *entropy* from Tele-Lens to identify pivot positions: along a CoT path, we select top-$k$ positions with the lowest final-answer entropy, as a proxy to indicate the confidence level of the entire path.

Accordingly, we conduct preliminary experiments with In-Domain LLM using the last Transformers layer: after the top-$k$ positions are selected, we obtain their average of final-answer entropy as a new uncertainty metric. The results, measured by standard AUROC, are presented in Table 2.

Comparing against conventional baselines over the full path, our top-$k$ selection strategy upon Tele-Lens signals achieves up to 9% absolute improvement upon the best baseline. Notably, the best estimation is obtained with $k = 5$ pivot tokens, demonstrating that latent signals from only a few positions can be a strong indicator of the whole path.

*Table 2.* Uncertainty estimation results (AUROC) with In-Domain LLM, using latent signals from final-answer probing via Tele-Lens (Section 3.1); values closer to 1 indicate better calibration. Using a subset of 5 positions along the CoT can better capture the uncertainty of the full path, with 9% substantial improvement over the best baseline. Full results across all tasks are shown in Table 6.

|  | GSM8K | Zebra | MMLU | GPQA | **Avg.** |
|---|---|---|---|---|---|
| Perplexity | 0.70 | 0.58 | 0.53 | 0.50 | 0.57 |
| Entropy | 0.72 | 0.60 | 0.52 | 0.50 | 0.58 |
| Self-Certainty | 0.76 | 0.67 | 0.53 | 0.51 | 0.60 |
| Tele-Lens (Top-5) | **0.87** | **0.77** | **0.73** | **0.56** | **0.69** |
| Tele-Lens (Top-10) | 0.81 | 0.75 | 0.72 | 0.56 | 0.68 |
| Tele-Lens (Top-20) | 0.82 | 0.67 | 0.65 | 0.51 | 0.63 |
| Tele-Lens (Top-50) | 0.78 | 0.69 | 0.56 | 0.47 | 0.64 |

**Latent Signals by General Metrics**    We next extend our validation to general scenarios without involving signals from a dedicated prober, lifting the constraint of a fixed answer space. We consider the three general metrics derived solely from predicted next-token logits over the model's vocabulary. For the generalizability of our findings, we conduct experiments with Off-the-Shelf LLMs, using both Qwen3-8B and Qwen3-32B. Specifically, we select the top-$k$ positions along a thinking path with the highest entropy / self-certainty, or with the lowest log-likelihood, respectively, representing the **most uncertain local steps** (the *shortest staves*). We use the average among these positions of each corresponding metric as the final estimation.

As shown in Table 3, for each LLM, applying the top-$k$ selection brings no negative impact using the selected $k$ values. The improvement is especially pronounced with Qwen3-32B: $k = 100$ consistently drives 3+% absolute improvement across all metrics, reaching up to 6%, thereby supporting the efficacy of our hypothesis.

Furthermore, we highlight the potential of exploiting richer latent signals and strategies for more effective calibration. On the one hand, the signals by Tele-Lens are highly informative, suggesting that additional training can be beneficial, consistent with prior findings that raw confidence metrics from LLMs alone may be sub-optimal (Kapoor et al., 2024). On the other hand, the spatial distributions of the selected *pivots* differ substantially between signals derived from Tele-Lens and those based on general metrics, as illustrated in Figure 8. These divergent distributions indicate that latent signals from multiple sources may provide complementary information, and their integration can be promising to further improve the identification of critical positions, leading to more robust uncertainty calibration.

### 3.2. CoT Necessity Estimation

We next conduct a study to estimate the necessity of CoT reasoning. In the broader context of improving model thinking efficiency, prior work has proposed various approaches

*Table 3.* Uncertainty estimation results (AUROC) with Qwen3-32B using the last Transformers layer, applying our top-$k$ strategy upon each general metric. Note that the average CoT length across inputs exceeds 7K tokens, while our simple strategy that selects top-100 positions is able to yield steady improvement. Full results across tasks with both 8B and 32B models are provided in Table 7.

| | GSM8K | MATH | MuSR | Zebra | CSQA | MMLU | QuALITY | GPQA | **Avg.** |
|---|---|---|---|---|---|---|---|---|---|
| Perplexity | 0.71 | **0.93** | 0.48 | 0.74 | 0.68 | 0.76 | 0.78 | 0.69 | 0.72 |
| w/ 100 Pivots | **0.81** | 0.92 | **0.50** | **0.90** | **0.74** | **0.81** | **0.82** | **0.73** | **0.78** |
| Entropy | 0.71 | **0.92** | 0.47 | 0.77 | 0.68 | 0.77 | 0.77 | 0.68 | 0.72 |
| w/ 100 Pivots | **0.81** | 0.70 | **0.49** | **0.90** | **0.74** | **0.83** | **0.82** | **0.74** | **0.75** |
| Self-Certainty | 0.45 | 0.82 | **0.47** | 0.92 | 0.51 | 0.67 | 0.64 | 0.68 | 0.65 |
| w/ 100 Pivots | **0.55** | **0.90** | **0.47** | **0.93** | **0.59** | **0.74** | **0.70** | **0.70** | **0.70** |

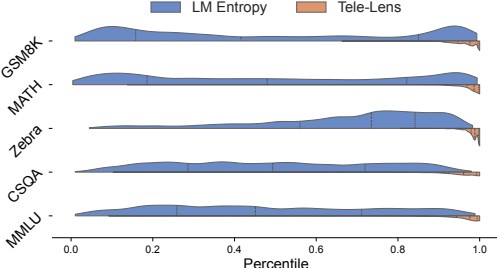

*Figure 8.* Spatial density distribution of selected *pivot* positions with In-Domain LLM along CoT paths. Using Tele-Lens, the selected positions tend to concentrate near CoT completion, whereas positions selected by general LM entropy typically are distributed across the entire CoT trajectory. Integrating multiple sources of latent signals may spur further improvement.

for reducing CoT overhead, such as early exiting (Yong et al., 2025; Yang et al., 2026) and CoT length steering (Li et al., 2026a). In this study, we investigate from a different perspective: assessing the input complexity to determine whether CoT reasoning is necessary in the first place.

Motivated by the observations in Figure 3, we leverage the early answer gist revealed by final-answer probing, where the probing accuracy may spike at the initial CoT positions for easier tasks. Our experiments show that these early signals can be effective recognizing whether a full CoT generation is required to accurately derive the final answer. By selectively bypassing CoT generation in non-essential cases, we can achieve a reduction in computational load with negligible performance degradation.

For each rollout specifically, we first generate its initial five CoT tokens and assess the normalized final-answer entropy $\bar{H}$ over logit distribution $\mathbf{p}$ across $C = 20$ probing classes:

$$\bar{H}(\mathbf{p}) = (-\sum_{i=1}^{C} p_i \log p_i)/\log C \qquad (6)$$

As $\bar{H}$ lies in the range $[0, 1]$, we adopt a threshold-based strategy: for initial positions, if any of their normalized entropy falls below a predefined threshold, representing a confident answer gist, we halt the corresponding CoT

*Table 4.* Evaluation results for CoT bypass, varying thresholds of normalized entropy from final-answer probing. The bypass ratio for each task is reported. **Avg.**: average bypass ratio; **Perf.**: average accuracy change. Full results are provided in Table 8.

| | Parity | CSQA | MMLU | GPQA | **Avg.** | **Perf.** |
|---|---|---|---|---|---|---|
| | | | In-Domain LLM | | | |
| Th=0.1 | 0% | 40.2% | 30.4% | 7% | 13.3% | -0.47 |
| Th=0.2 | 0% | 65% | 45% | 12% | 21.6% | -1.42 |
| | | Off-the-Shelf LLM (Qwen3-32B) | | | | |
| Th=0.1 | 0% | 16.2% | 12.4% | 1.2% | 2.8% | -0.03 |
| Th=0.2 | 0% | 28.8% | 20.2% | 3.2% | 6.2% | -0.37 |

generation and directly output the answer by disabling the LLM's thinking mode, bypassing a full generation. The evaluation results are reported in Table 4.

With the threshold set to 0.1, our objective is robustly accomplished for both In-Domain and Off-the-Shelf LLMs: the aforementioned heuristic automatically recognizes inputs for which CoT is necessary to derive the final answer, such as Parity, while bypassing CoT generation on easier tasks, such as CSQA. For instance, Qwen3-32B attains 16.2% / 12.4% thinking reduction for CSQA / MMLU almost "for free", with only 0.03% overall accuracy degradation.

As our necessity estimation relies on a fixed answer space to obtain the metric, we present it primarily as a *proof-of-concept*. Nevertheless, we underscore the significance of exploiting such useful latent signals, which can benefit various scenarios, such as to facilitate model training (Huang et al., 2025) and CoT compression (Singh & Hakkani-Tür, 2026; Li et al., 2026b) and steering (Li et al., 2026a). We provide further discussions and Related Works in Appendix E.

## 4. Conclusion

In this work, we investigate the internal planning capacity of LLMs and uncover a myopic planning horizon during CoT generation. For analysis, we design a series of probing experiments using our proposed method, Tele-Lens, and we highlight the exploitation of such latent signals, demonstrating on both CoT uncertainty and necessity estimation.

# Impact Statement

This paper presents work whose goal is to advance the understanding of internal dynamics of Large Language Models, in particular the latent planning horizon and the according utilization. There may be potential societal consequences of this work, none which we feel must be specifically highlighted here.

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

# A. Tasks and Datasets

As described in Section 2.2, our probing experiments span 12 diverse tasks of different types for a comprehensive view of empirical insights. This section further provides concrete examples, data processing details and statistics.

## A.1. Task Examples

---

**Task Example for: Parity**

```
[Settings]:
  • Sequence length:  41
  • Target digit to count:  2
  • Answer:  even
```
- - - - - - - - - - - - - - - - - - - - - - - - - - - -

Determine whether the number of "2" in the following digit sequence is even or odd; please output only your decision by either "even" or "odd".

91223822122515222430601862928242722242251

---

**Task Example for: Cycle**

```
[Settings]:
  • Number of edges:  16
  • Answer:  NO
```
- - - - - - - - - - - - - - - - - - - - - - - - - - - -

**Task**
Given the following directed graph represented as a list of edges (from_vertex → to_vertex), along with two target vertices, you need to determine whether there exists a path from the first target vertex to the second.

**Edges**
v453 → v561
v666 → v34
v34 → v791
v791 → v17
v416 → v0
v658 → v666
v0 → v74
v254 → v427
v427 → v520
v561 → v254
v74 → v453
v520 → v416
v664 → v464
v17 → v664
v640 → v658
v464 → v640

**Target**
v_34, v_561

**Output**
Please output only "YES" if a path exists, or "NO" if it does not.

---

**Task Example for: Subsum**

```
[Settings]:
  • Sequence length:  29
  • Max subsequence sum:  84
  • Answer:  4
```
- - - - - - - - - - - - - - - - - - - - - - - - - - - -

Given the following sequence of numbers, determine the least significant digit of the maximum sum of its subsequences, such that no two numbers in the subsequence are adjacent in the original sequence. Please output only the according least significant digit directly.

[2, 4, 6, 6, 1, 8, 5, 5, 4, 6, 6, 6, 6, 8, 1, 8, 9, 1, 9, 9, 4, 1, 9, 5, 4, 2, 4, 3, 2]

---

**Task Example for: GSM8K (Multi-Choice)**

```
[Settings]:
  • Answer:  D
```
- - - - - - - - - - - - - - - - - - - - - - - - - - - -

**Task**
Given the following problem along with its options, determine the best option as the answer. Please only output your selected answer option by the letter (e.g., A, B, C).

**Problem**
Rob, Royce, and Pedro are contractors getting ready to put a new roof on three homes. If the three homes will need 250 cases of shingles, with the first house needing 1/2 of the second, and the third needing double the first. How many cases of shingles will the third house need?

Options:
A. 125
B. 200
C. 50
D. 100
E. 83

---

**Task Example for: MATH (Multi-Choice)**

```
[Settings]:
  • Answer:  B
```
- - - - - - - - - - - - - - - - - - - - - - - - - - - -

**Task**
Given the following problem along with its options, determine the best option as the answer. Please only output your selected answer option by the letter (e.g., A, B, C).

**Problem**
If each point of the circle $x^2 + y^2 = 25$ is reflected in the point $(4, 1)$, the set of image points satisfies the equation

$$x^2 + ay^2 + bx + cy + d = 0.$$

Compute the ordered quadruple $(a, b, c, d)$ of real numbers.

Options:
A. (1,16,4,43)

---

B. (1,-16,-4,43)
C. (1,-8,-2,17)
D. (1,-16,4,43)
E. (1,-16,-4,-43)

---

### Task Example for: AIME (Multi-Choice)

```
[Settings]:
  • Answer:  B
```
--------------------------------------------

**Task**
Given the following problem along with its options, determine the best option as the answer. Please only output your selected answer option by the letter (e.g., A, B, C).

**Problem**
The set of points in 3-dimensional coordinate space that lie in the plane $x + y + z = 75$ whose coordinates satisfy the inequalities $x - yz < y - zx < z - xy$ forms three disjoint convex regions. Exactly one of those regions has finite area. The area of this finite region can be expressed in the form $a\sqrt{b}$, where $a$ and $b$ are positive integers and $b$ is not divisible by the square of any prime. Find $a + b$.

Options:
A. 524
B. 510
C. 498
D. 504
E. 496

---

### Task Example for: MuSR

```
[Settings]:
  • Answer:  A
```
--------------------------------------------

**Task**
Given the following article, along with a related question and its answer options, please determine the best answer option for this question.

**Article**
In my latest tenure at a bustling educational institution, three staff members, Emily, Robert, and Alice, consistently caught my attention amidst the sea of educators and support personnel. As the school manager, my role was to distribute tasks, specifically Teaching and Admin work, in a way that capitalized on each individual's unique strengths, thereby streamlining the school's operations. These assignments, as crucial as they were intricate, were akin to the individual notes in a symphony, each playing a vital role in the harmony of the institution.
Alice was a unique blend of complexities, as I carefully observed her interactions with the staff. Her proclivity for administrative tasks was evident, a much-needed quality in the heaving sea of paperwork the school generated. Alice often took the responsibility of capturing the minutes during our staff meetings and backed up Robert's teachings with her painstaking administrative work...
...

---

**Question**
Given the story, how would you uniquely allocate each person to make sure both tasks are accomplished efficiently?
A. Teaching: Emily, Admin work: Alice and Robert
B. Teaching: Alice, Admin work: Emily and Robert
C. Teaching: Robert, Admin work: Alice and Emily

**Output**
Please only output your selected answer option by "A/B/C/...".

---

### Task Example for: Zebra

```
[Settings]:
  • Answer:  A
```
--------------------------------------------

**Task**
Given the following problem along with its options, determine the best option as the answer. Please only output your selected answer option by the letter (e.g., A, B, C).

**Problem**
There are 5 houses, numbered 1 to 5 from left to right, as seen from across the street. Each house is occupied by a different person. Each house has a unique attribute for each of the following characteristics:
- Each person has a unique name: Peter, Alice, Arnold, Bob, Eric
- Each person has a unique hobby: photography, cooking, knitting, gardening, painting
- Each person has a unique favorite drink: root beer, milk, water, coffee, tea

Rules:
1. Eric is the coffee drinker.
2. The tea drinker is the person who paints as a hobby.
3. The person who enjoys knitting is not in the fourth house.
4. Peter is not in the fourth house.
5. Eric is somewhere to the right of the root beer lover.
6. Arnold is the person who loves cooking.
7. The one who only drinks water is somewhere to the right of the person who enjoys gardening.
8. There is one house between Bob and the person who paints as a hobby.
9. The person who enjoys gardening is directly left of the root beer lover.
10. The photography enthusiast is the one who only drinks water.

Question:
What is Drink of the person who lives in House 1?
A. milk
B. tea
C. root beer
D. coffee
E. water

---

## Task Example for: CSQA

```
[Settings]:
  • Answer:  B
```
-----------------------------------------

**Task**
Given the following commonsense question, please determine its best answer option.

**Question**
The drought was dangerous for the trees, they were more likely to what?
A. fire
B. burn
C. covered in snow
D. wall in
E. grow tall

**Output**
Please only output your selected answer option by "A/B/C/...".

## Task Example for: MMLU

```
[Settings]:
  • Answer:  D
```
-----------------------------------------

**Task**
Given the following question and its options, determine the best option as the answer. Please only output your selected answer option by "A/B/C/D".

**Question**
A manager's competitor sent a defamatory letter to the manager accusing him of professional incompetence and calling him one of the worst businessmen in town. It was addressed to the manager. He read it, put it in a private drawer, and did not read it again. Later, he tried to sue the competitor for defamation as a result of the letter. Will the court likely grant the defendant's motion to dismiss, and on what grounds? Base your answer on the common law definition of defamation.

A. No, it will not dismiss because a plaintiff in a defamatory action has an absolute right to a jury trial to prove defamation.
B. Yes, it will dismiss on the basis that the language is not damaging to the manager's reputation.
C. No, it will not dismiss because the circumstances show that all of the elements of defamation are all present.
D. Yes, it will dismiss on the basis that the publication is made to the manager alone.

## Task Example for: QuALITY

```
[Settings]:
  • Answer:  B
```
-----------------------------------------

**Task**
Given the following snippets from an article or a story,

along with a related question and its answer options, you need to determine the best option based on the information provided by these snippets.
These snippets may not necessarily always contain the supported information to answer the question; in that case try to give a best guess.

**Snippets**
For his earlier errors, Coleman had first received a suspended sentence, then two terminal sentences to be fixed by the warden. My predecessors had given him first a few weeks, then a few months of sleep in Dreamland. Coleman's eyes didn't frighten me; I focused right on the pupils. "That was a pretty foul trick, Councilman. Did you hope to somehow frighten me out of executing this sentence by what you told me this morning?"
I couldn't follow his reasoning. Just how making me think my life was only a Dream such as I imposed on my own prisoners could help him, I couldn't see...
...

**Question**
What were Coleman's motivations in visiting the warden?
A. Providing the warden with his annual raise announcement
B. Scaring him into believing his life was a dream
C. Gathering information to bring down the warden's compound
D. Persuading the warden to step down from his position

**Output**
Please only output your selected answer option by "A/B/C/D".

## Task Example for: GPQA

```
[Settings]:
  • Answer:  B
```
-----------------------------------------

**Task**
Given the following exam question, please determine its best answer option.

**Question**
Observations of structures located at a distance of about 2.1 gigaparsecs (2.1 Gpc) are being carried out. The detected absorption line energy equivalent is about 3.9 micro electron volts ($3.9 * 10^{-6}$ eV).

What is most likely to be observed with this absorption line in the Milky Way?
A. Warm atomic interstellar medium.
B. Cold atomic interstellar medium.
C. Warm molecular interstellar medium.
D. Cold molecular interstellar medium.

**Output**
Please only output your selected answer option by "A/B/C/D".

## A.2. Dataset Descriptions

A brief description of each existing dataset adopted in our experiments is provided below.

**Implicit Compositional Tasks**  Three mathematical tasks and two logical reasoning tasks are included:

- **GSM8K**: a dataset focusing on middle school-level problems of various difficulties (Cobbe et al., 2021).
- **MATH**: a dataset introduced by Hendrycks et al. (2021b) with high school competition-level math problems. We follow the MATH-500 test split by Lightman et al. (2024).
- **AIME**: 30 math competition problems from AIME'25 (2025 American Invitational Mathematics Examination) (AIME, 2025).
- **MuSR**: a Multistep Soft Reasoning dataset to evaluate logical deduction with natural language rules over long, text-based narratives (Sprague et al., 2024).
- **Zebra**: the benchmark ZebraLogic (Lin et al., 2025) designed to evaluate symbolic reasoning and constraint satisfaction abilities within a natural language context.

**Knowledge and Semantic Tasks**  Four knowledge-intensive benchmarks focusing on semantic understanding rather than explicit reasoning are included:

- **CSQA**: CommonsenseQA (Talmor et al., 2019), targeting commonsense reasoning based on world knowledge.
- **MMLU**: a broad-spectrum dataset to evaluate knowledge from 57 subjects, covering various STEM and social science domains (Hendrycks et al., 2021a).
- **QuALITY**: a narrative question answering dataset (Pang et al., 2022). To reduce computational overhead, we frame the context in the form of relevant snippets retrieved via dense retrieval (a RAG setting with a max 2K context length), rather than using full documents.
- **GPQA**: a challenging dataset designed to test expert-level knowledge of multiple domains, such as biology, physics, chemistry, etc. (Rein et al., 2024).

## A.3. Data Preparation For Existing Datasets

Among the 12 tasks used in our probing experiments, three mathematics tasks (MATH, GSM8K, and AIME) are originally evaluated via free-form generation, without a fixed answer space. To enable final-answer probing, we use the following prompt to convert each problem into a multiple-choice format using GPT-4.1.

---

**Prompt for Multi-Choice Conversion**

**Task**
Given the following problem and its correct answer solution, you need to generate four plausible but incorrect answer options, which will serve as misleading distractors

---

to construct multiple-choice questions.

**Problem**
{problem}

**Solution** [Optional]
{solution}

**Correct Answer**
{answer}

**Output**
Please first think about four misleading wrong answer options, starting with "### Think".
Then, starting with "### Options", provide each option per line, where each line is directly the according answer option in a similar format of the correct answer, without adding any prefix or explanation.

---

For other existing datasets originally in a multiple-choice format, we also shuffle the order of options for each question, intended to mitigate potential memorization effects and positional bias by LLMs.

With all 12 tasks having a fixed answer space, the label set for the final-answer probing consists of 20 tokens in total:

$$\left\{ \begin{array}{l} \text{A, B, C, D, E, F, YES, NO, even, odd,} \\ 0, 1, 2, 3, 4, 5, 6, 7, 8, 9 \end{array} \right\}$$

## A.4. Data Generation and Sampling

The data generation process for the three explicit compositional tasks is fully controllable. For probing, we generate problems and their corresponding labels for each task using the following procedure.

**Parity**  We generate random digit sequences from length 5 to 100. For each sequence, the target digit to count is randomly selected from $\{1, 2, 7, 8\}$. The label is then determined by the parity of its count.

**Cycle**  For each input, we first set the number of edges, an even number randomly determined from 4 to 100. Two instances are yielded at each time, one forming a full cycle using all edges, and another forming two equal-sized cycles each using half of the edges. We then randomly assign vertex names from a pool of 1000 candidates to each cycle, ensuring diversity in both vertex identities and edge orderings. For the target vertices to determine whether there exists a path in between, we randomly select two vertices in the first case (there is always a path due to cycling), and randomly select one vertex from each cycle for the second case (no path exists).

**Subsum**  We generate random lists of integers (each in the range from 1 to 9), with lengths ranging from 2 to 50. The

labels are obtained by applying the dynamic programming solution to each list.

With our data generation process, the resulting labels for each task are evenly distributed. Our code for data generation and the resulting datasets will be publicly released.

**Data Sampling** For other tasks, we sample from their original test sets, such as our test split always consists of problems drawn from the original test sets. For our train/dev splits, we keep sampling remaining problems from the test sets. If a dataset does not contain a sufficient number of test instances, we additionally sample from their original training and dev sets, if they are available.

Note that AIME'25 contains only 30 problems in total. Following the above procedure, all 30 problems are included in our test split, with no problems added to our train or dev split for AIME.

### A.5. Dataset Construction and Statistics

With our train/dev/test splits in place, we perform inference by each of the two LLM backbones, collecting their corresponding rollouts and hidden states. We retain hidden states of all tokens along CoT trajectories within a maximum length of 16,384 for the test split. For the train/dev splits, to maintain a manageable storage cost, we keep hidden states of sampled 5% / 10% CoT tokens for Off-the-Shelf LLM and In-Domain LLM respectively.

The resulting train/dev/test dataset has 2.4M / 81K / 11M hidden states for Off-the-Shelf LLM, and 2.5M / 57K / 2.7M hidden states for In-Domain LLM, respectively (for each Transformers layer).

Each hidden state is labeled according to the corresponding rollout outcomes for each teleological dimension, e.g. the tokens ID for the $i$-th subsequent token, the token ID of the final predicted answer, the total CoT length, etc. Given the ample number of instances in our dataset, our Tele-Lens adapters can automatically learn latent features that discriminate among different labels.

### A.6. Hyperparameters

The hidden size is $d = 5120$ for Off-the-Shelf LLM (Qwen3-32B), and $d = 3584$ for In-Domain LLM that is trained upon Qwen2.5-7B-Instruct. We use $r = 256$ for both models, which is shown sufficient by prior works on probing (Dong et al., 2025). For training, we adopt learning rate $1 \times 10^{-3}$, batch size 8000, a linear decay learning rate schedule, and early stopping on dev set, with approximately 5000 max training steps; we do not enable weight decay or warmup period. The training is conducted on Nvidia V100 GPUs, and only the parameters of Tele-Lens are updated; the LM head is kept frozen during training.

## B. In-Domain LLM

### B.1. Training Details

As described in Section 2.3, we conduct reinforcement learning with GRPO (Shao et al., 2024) on Qwen2.5-7B-Instruct to obtain an In-domain LLM on the 12 tasks. During training, we add the format instruction in the LLM system prompt, and use a reward signal based solely on format validation and answer correctness (each with score 1). The training set comprises 48K problems, sampled from original training sets of those existing datasets, as well as auto-generated problems for explicit compositional tasks.

> **System Prompt for: In-Domain LLM**
>
> ```
> You are a helpful assistant.  Now the user asks
> you to solve a reasoning problem.  You need to
> first think about the solving process in the
> mind and then provide the user with the answer.
> The thinking process is enclosed within <think>
> </think> tags, i.e., <think> thinking process
> here </think> final answer.
> ```

For training, we use a rollout size 16, batch size 320 with mini batch size 80, capping the max response length as 4096. We adopt a cosine learning rate schedule with initial learning rate $1 \times 10^{-6}$ and 10 warmup steps. Following DAPO (Yu et al., 2025), we adopt the clip-higher strategy to encourage exploration, setting the upper clip ratio as 0.3 and the lower clip ratio as 0.2. The training converges within 800 steps, among which Parity is the slowest to converge.

### B.2. Evaluation

Evaluation of our adopted LLM backbones across 12 tasks, including our In-Domain LLM trained by reinforcement learning, are provided in Table 5.

### B.3. In-Domain CoT Examples

As shown in Table 5, our In-Domain LLM produces much shorter CoT trajectories, indicating more stable and decisive reasoning behavior. For qualitative comparison, we provide examples for Parity using both Off-the-Shelf LLM (Qwen3-32B) and In-Domain LLM, shown in Figure 9 and Figure 10. Another example for Cycle is shown in Figure 11.

## C. Probing Results

### C.1. Collection of Full Results

Section 2.5 reports the empirical results of our probing settings. Due to the limited space in the main pages, we present figures and tables for full results across layers and tasks, listed as below.

*Table 5.* Accuracy of LLM backbones on the 12 tasks spanning three categories described in Section 2.2, along with CoT length (measured in number of characters), averaged from 5 repeated runs. For the off-the-shelf Qwen3 LLMs, we evaluate two settings, with thinking mode disabled (w/o CoT) or enabled (w/ CoT), respectively. Our In-Domain LLM is trained by GRPO upon Qwen2.5-7B-Instruct (details in Section 2.3), which is one generation behind the Qwen3 series, thus its performance may lag behind on certain datasets. Despite this, it achieves the best performance on three compositional tasks while producing **substantially shorter CoT trajectories**. Note that in our evaluation, the maximum CoT length is capped at 32,768 tokens; any response exceeding this limit is considered incorrect. Result discussions are addressed in Section 2.5.

| | Explicit Comp. | | | Implicit Comp. | | | | | Knowledge & Semantics | | | | |
|---|---|---|---|---|---|---|---|---|---|---|---|---|---|
| | Parity | Cycle | Subsum | GSM8K | MATH | AIME | MuSR | Zebra | CSQA | MMLU | QuALITY | GPQA | **Avg.** |
| Random | 50 | 50 | 10 | 20 | 20 | 20 | 20 | 16.7 | 20 | 25 | 25 | 20 | 24.7 |
| Qwen3-8B (w/o CoT) | 49.6 | 45.4 | 0.6 | 28.0 | 26.9 | 3.3 | 58.4 | 43.8 | 76.8 | 70.6 | 87.0 | 35.0 | 43.8 |
| Qwen3-8B (w/ CoT) | 76.5 | 92.8 | 89.1 | 97.0 | **98.3** | **76.8** | 65.6 | 87.9 | 79.2 | 85.3 | 91.5 | 55.5 | 83.0 |
| *CoT Length* | *10550* | *16215* | *23503* | *2848* | *9873* | *44022* | *7904* | *22714* | *3221* | *4658* | *3523* | *20348* | *14114.9* |
| Qwen3-32B (w/o CoT) | 46.8 | 45.4 | 8.4 | 49.0 | 39.3 | 33.3 | 55.8 | 50.6 | 77.6 | 78.8 | 92.6 | 42.6 | 51.7 |
| Qwen3-32B (w/ CoT) | 80.9 | 96.6 | **94.7** | **97.9** | 97.7 | 76.7 | 65.5 | 90.2 | **82.3** | **91.4** | **97.3** | **63.8** | **86.2** |
| *CoT Length* | *6390* | *11951* | *13758* | *1678* | *6865* | *39085* | *4835* | *21306* | *2090* | *2920* | *2089* | *14793* | *10646.6* |
| In-Domain LLM (7B) | **98.3** | **97.2** | 87.6 | 94.4 | 87.3 | 56.7 | 53.8 | **94.9** | 78.4 | 78.5 | 89.9 | 36.4 | 79.4 |
| *CoT Length* | *1699* | *972* | *2718* | *966* | *1268* | *1521* | *1013* | *1479* | *749* | *1247* | *628* | *1513* | *1314.6* |

<think>
Okay, let's see. I need to figure out if the number of 7s in this sequence is even or odd. The sequence given is 71270777577717. Hmm, first step is probably to count how many 7s there are.

Alright, let me start from the beginning and go through each digit one by one. Let's write down the number again to make sure I don't miss any digits: 7, 1, 2, 7, 0, 7, 7, 7, 5, 7, 7, 7, 1, 7. Wait, let me check that again. The original number is 71270777577717. Breaking it down:

Position 1: 7
Position 2: 1
Position 3: 2
...
Position 12: 7
Position 13: 1
Position 14: 7

Okay, that's 14 digits total. Now, let me count the 7s. Let's go step by step:

1. First digit is 7 → count = 1
2. Second is 1 → not a 7 → count remains 1
3. Third is 2 → not a 7 → count remains 1
...
12. Twelfth is 7 → count = 8
13. Thirteenth is 1 → no → count remains 8
14. Fourteenth is 7 → count = 9

Wait, so according to this, there are 9 sevens. But 9 is an odd number. So the answer should be "odd". Let me double-check my count to make sure I didn't skip a digit or miscount.

Let me list all the digits with their positions and whether they are 7:

1: 7 → yes
2: 1 → no
3: 2 → no
...
12:7 → yes
13:1 → no
14:7 → yes

Now, let's count the "yes" responses. Let's list the positions where it's a 7: 1,4,6,7,8,10,11,12,14. That's 9 positions. Yep, 9 is odd. So the number of 7s is odd. Therefore, the answer should be "odd".
</think>
odd

*Figure 9.* Example of Parity Response with Off-the-Shelf LLM (Qwen3-32B). Full evaluation is discussed in Section 2.5.

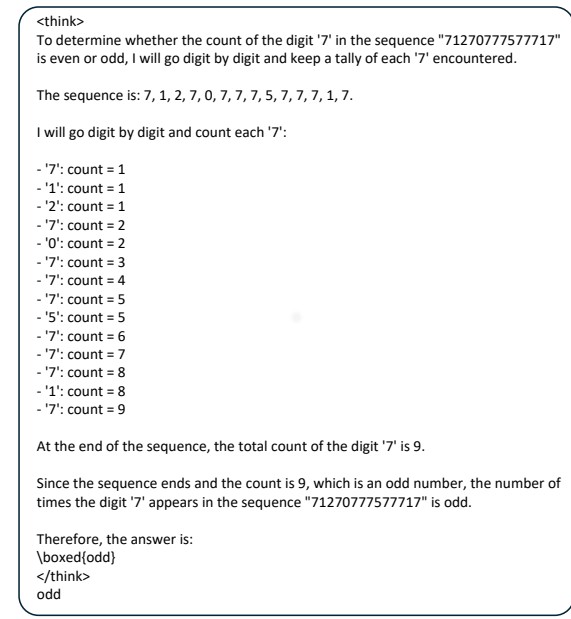

<think>
To determine whether the count of the digit '7' in the sequence "71270777577717" is even or odd, I will go digit by digit and keep a tally of each '7' encountered.

The sequence is: 7, 1, 2, 7, 0, 7, 7, 7, 5, 7, 7, 7, 1, 7.

I will go digit by digit and count each '7':

- '7': count = 1
- '1': count = 1
- '2': count = 1
- '7': count = 2
- '0': count = 2
- '7': count = 3
- '7': count = 4
- '7': count = 5
- '5': count = 5
- '7': count = 6
- '7': count = 7
- '7': count = 8
- '1': count = 8
- '7': count = 9

At the end of the sequence, the total count of the digit '7' is 9.

Since the sequence ends and the count is 9, which is an odd number, the number of times the digit '7' appears in the sequence "71270777577717" is odd.

Therefore, the answer is:
\boxed{odd}
</think>
odd

*Figure 10.* Example of Parity Response with our In-Domain LLM trained via GRPO. The resulting reasoning trajectory is much shorter with predictable patterns, as discussed in Section 2.5.2.

- Probing for **Final Answers**
  - Figure 13: the probing accuracy with In-Domain LLM at the beginning of CoT.
  - Figure 14: the probing accuracy with Off-the-Shelf LLM at the beginning of CoT.
  - Figure 15: examples of probing accuracy dynamics along CoT trajectories with Off-the-Shelf LLM.
  - Figure 17&18: averaged probing accuracy along CoT trajectories with Off-the-Shelf LLM.
  - Figure 19: task accuracy comparison for Off-the-Shelf LLM under settings that include think-

```
<think>
To determine if there is a path from vertex v_815 to vertex v_109, I will trace the
directed edges starting from v_815 and see if I can reach v_109.

Starting from v_815:
1. v_815 -> v_402
2. v_402 -> v_361
3. v_361 -> v_963
4. v_963 -> v_353
5. v_353 -> v_242
…
27. v_346 -> v_131
28. v_131 -> v_398
29. v_398 -> v_39
30. v_39 -> v_728
31. v_728 -> v_109

Since we reached v_109 from v_815, there exists a path.

Therefore, the answer is "YES".
</think>
YES
```

*Figure 11.* Example of Cycle Response with our In-Domain LLM. The length of the reasoning trajectory is in proportional to the length of the path/cycle between two vertices, but not to the number of input edges. Without the latter heuristics, LLM is not able to reliably predict the total reasoning length at the initial stage of CoT, as discussed in Section 2.5.3.

ing mode, non-thinking mode, early final-answer planning, random guess.
   – Figure 20: comparison similar to Figure 19 for In-Domain LLM.
- Probing for **Subsequent Tokens**
   – Figure 21: Top-5 accuracy for subsequent token prediction with In-Domain LLM, up to the 8th following token.
   – Figure 22: evaluation similar to Figure 21 with Off-the-Shelf LLM.
- Probing for **Global Steps**
   – Figure 23: heatmap of reasoning length probing with In-Domain LLM.
   – Figure 24: heatmap of reasoning length probing with Off-the-Shelf LLM.

## C.2. More Myopic Planning Illustrations

Figure 16 illustrates the final-answer planning dynamics on tasks beyond explicit compositional reasoning. They also exhibit a *myopic* planning horizon, with high-confidence probing positions appearing sparsely. Similar to explicit compositional tasks, such positions tend to emerge near CoT completion in math and logical reasoning tasks as well.

## D. Leveraging CoT Dynamics

### Results

- Figure 12 presents the density distribution of LM entropy for next-token prediction.
- Table 6 and 7 show the full evaluation results for uncertainty estimation, using our top-$k$ pivot selection strategy described in Section 3.1.

- Table 8 presents the full evaluation results of CoT bypass described in Section 3.2.

## E. Related Works and Discussions

There can be many implications brought by the *myopic* planning horizon uncovered in this work. Since the model cannot plan the end from the beginning, it must initiate the dynamic reasoning as an necessary act of state searching and exploration. Therefore, explicit planning within CoT can be important, as empirically validated by recent works (Wang et al., 2023; 2025a; 2026). The exploitation of latent signals from CoT dynamics can be significant to various LLM characteristics and applications. For instance, recent works have investigated to utilize latent signals to compress CoT (Zhang et al., 2025; Li et al., 2026b), steer model behavior (Sheng et al., 2025; Li et al., 2026a), perform early stop of CoT (Afzal et al., 2025; Singh & Hakkani-Tür, 2026; Li et al., 2026b), and improve model training (Huang et al., 2025). Thus, we believe understanding the underlying CoT dynamics is significant to address in research.

To understand the internal states of LLMs, prior works have conducted probing studies on Transformers' hidden states to address truthful responses (Azaria & Mitchell, 2023; Liu et al., 2024; Gottesman & Geva, 2024; Chen et al., 2026), assess world knowledge representation (Patel & Pavlick, 2022; Li et al., 2023) or the global planning prior to CoT generation (Dong et al., 2025). In terms of probing methods, in addition to training probers on LLM hidden states as in (Dong et al., 2025), other methods have been proposed as well, such as Patchscopes (Ghandeharioun et al., 2024). In this work, we opt in probers for the flexibility predicting various targets, as we have three probing dimensions described in Section 2.

Apart from probing, prior works have studied CoT dynamics from different aspects. Wang et al. (2025b) finds that only about 20% tokens are of high entropy. Bigelow et al. (2025) proposes a sampling-based method for pivot token identification. Ton et al. (2025) proposes a methodology to quantify information gain at each CoT step. (Shao et al., 2025) proposes CoT Several works have also identified that CoT could bring negative impact in certain scenarios (Sprague et al., 2025; Liu et al., 2025).

In additional to analyses related to CoT dynamics, studies on Transformers' learnability and expressibility have highlighted the functional necessity of CoT on certain problems. Several works have focused on the theoretical limitation of Transformers, where it fails to perform soft multi-step reasoning within one step (Bhattamishra et al., 2023; Merrill & Sabharwal, 2023; Li et al., 2024), and only intermediate CoT steps can derive length generalization (Anil et al., 2022; Xiao & Liu, 2025) and compositional reasoning (Wies et al.,

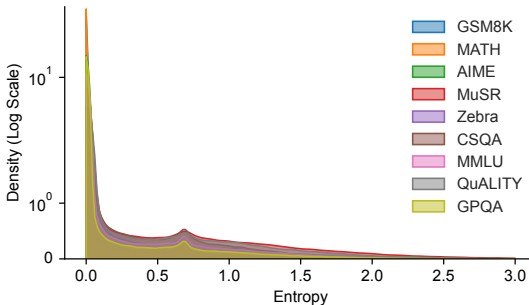

*Figure 12.* Density distribution of LM entropy for next-token prediction steps per task. As depicted, most tokens exhibit low entropy, reflecting confident local transitions (Section 3.1).

2023; Abbe et al., 2024; Zubic et al., 2025), making CoT indispensable especially for compositional problems. Our experiments in this work generally align with those findings.

To the best of our knowledge, this work is the first to focus explicitly on the latent planning horizon and its effective utilization, offering a unified perspective on prior work from complementary angles. We also call for attention on the identification and exploitation of more such hidden yet valuable latent signals to further deepen our understanding of CoT synergy.

*Table 6.* Uncertainty estimation results (AUROC) with In-Domain LLM, using latent signals from final-answer probing via Tele-Lens. Note that we exclude auto-generated tasks (Parity, Cycle, Subsum), as their uncertainty is already artificially correlated with input length. Result discussions are provided near Table 2.

|                     | GSM8K | MATH | AIME | MuSR | Zebra | CSQA | MMLU | QuALITY | GPQA | Avg. |
|---------------------|-------|------|------|------|-------|------|------|---------|------|------|
| Perplexity          | 0.70  | 0.64 | 0.49 | 0.50 | 0.58  | 0.57 | 0.53 | 0.64    | 0.50 | 0.57 |
| Entropy             | 0.72  | 0.65 | 0.49 | 0.51 | 0.60  | 0.58 | 0.52 | 0.63    | 0.50 | 0.58 |
| Self-Certainty      | 0.76  | 0.71 | 0.51 | **0.52** | 0.67 | 0.58 | 0.53 | 0.64 | 0.51 | 0.60 |
| Tele-Lens (Top-5)   | **0.87** | **0.81** | **0.65** | 0.49 | **0.77** | **0.64** | **0.73** | **0.72** | **0.56** | **0.69** |
| Tele-Lens (Top-10)  | 0.81  | **0.81** | 0.63 | 0.49 | 0.75 | 0.63 | 0.72 | 0.70 | 0.56 | 0.68 |
| Tele-Lens (Top-20)  | 0.82  | 0.79 | 0.56 | 0.49 | 0.67 | 0.52 | 0.65 | 0.66 | 0.51 | 0.63 |
| Tele-Lens (Top-50)  | 0.78  | 0.73 | 0.49 | 0.50 | 0.69 | 0.40 | 0.56 | 0.56 | 0.47 | 0.64 |

*Table 7.* Uncertainty estimation results (AUROC) with Off-the-Shelf LLMs, using our top-$k$ strategy upon each general metric. Note that we exclude auto-generated tasks (Parity, Cycle, Subsum), as their uncertainty is already artificially correlated with input length. We also exclude AIME, as there are not enough negative instances from both models. Our top-$k$ strategy brings no negative impact, and particularly drives consistent improvement with Qwen3-32B. Result discussions are provided near Table 3.

|           |                | GSM8K | MATH | MuSR | Zebra | CSQA | MMLU | QuALITY | GPQA | Avg. |
|-----------|----------------|-------|------|------|-------|------|------|---------|------|------|
| Qwen3-8B  | Perplexity     | 0.85  | 0.84 | 0.54 | 0.73  | **0.77** | **0.84** | 0.85 | **0.72** | 0.77 |
|           | + 10 Pivots    | 0.82  | 0.82 | 0.52 | 0.79  | 0.73 | 0.79 | 0.81 | 0.61 | 0.74 |
|           | + 100 Pivots   | 0.87  | **0.87** | 0.58 | 0.89  | 0.76 | 0.82 | 0.85 | 0.67 | 0.79 |
|           | + 1000 Pivots  | **0.88** | **0.87** | **0.59** | **0.94** | **0.77** | 0.82 | **0.86** | 0.70 | **0.80** |
|           | Entropy        | 0.85  | 0.83 | 0.55 | 0.76  | **0.76** | 0.85 | 0.85 | **0.73** | 0.77 |
|           | + 10 Pivots    | 0.87  | 0.84 | 0.54 | 0.93  | **0.76** | 0.82 | 0.84 | 0.70 | 0.79 |
|           | + 100 Pivots   | **0.88** | 0.85 | 0.57 | **0.94** | **0.76** | 0.82 | **0.86** | 0.72 | **0.80** |
|           | + 1000 Pivots  | **0.88** | 0.86 | **0.59** | **0.94** | **0.76** | 0.82 | **0.86** | 0.71 | **0.80** |
|           | Self-Certainty | 0.87  | **0.92** | 0.58 | **0.96** | **0.77** | **0.84** | 0.86 | 0.72 | **0.82** |
|           | + 10 Pivots    | 0.86  | 0.91 | 0.55 | 0.95  | **0.77** | 0.83 | 0.85 | **0.73** | 0.81 |
|           | + 100 Pivots   | **0.88** | 0.90 | 0.58 | **0.96** | 0.76 | 0.83 | **0.87** | **0.73** | **0.82** |
|           | + 1000 Pivots  | **0.88** | 0.91 | **0.59** | **0.96** | 0.76 | 0.83 | **0.87** | 0.72 | **0.82** |
| Qwen3-32B | Perplexity     | 0.71  | 0.93 | 0.48 | 0.74  | 0.68 | 0.76 | 0.78 | 0.69 | 0.72 |
|           | + 10 Pivots    | 0.74  | 0.79 | 0.48 | 0.75  | 0.69 | 0.76 | 0.79 | 0.71 | 0.71 |
|           | + 100 Pivots   | **0.81** | 0.92 | **0.50** | 0.90  | **0.74** | **0.81** | **0.82** | 0.73 | **0.78** |
|           | + 1000 Pivots  | 0.72  | **0.95** | 0.49 | **0.91** | 0.71 | 0.80 | 0.80 | **0.74** | 0.76 |
|           | Entropy        | 0.71  | **0.92** | 0.47 | 0.77  | 0.68 | 0.77 | 0.77 | 0.68 | 0.72 |
|           | + 10 Pivots    | 0.78  | 0.69 | 0.48 | 0.87  | 0.71 | 0.79 | 0.80 | 0.73 | 0.73 |
|           | + 100 Pivots   | **0.81** | 0.70 | **0.49** | 0.90  | **0.74** | **0.83** | **0.82** | **0.74** | **0.75** |
|           | + 1000 Pivots  | 0.71  | 0.87 | 0.48 | **0.91** | 0.71 | 0.81 | 0.79 | 0.73 | **0.75** |
|           | Self-Certainty | 0.45  | 0.82 | 0.47 | 0.92  | 0.51 | 0.67 | 0.64 | 0.68 | 0.65 |
|           | + 10 Pivots    | 0.53  | 0.89 | 0.47 | 0.91  | 0.57 | 0.71 | 0.67 | 0.69 | 0.68 |
|           | + 100 Pivots   | **0.55** | 0.90 | 0.47 | **0.93** | **0.59** | **0.74** | **0.70** | 0.70 | **0.70** |
|           | + 1000 Pivots  | 0.52  | **0.91** | **0.48** | **0.93** | 0.54 | **0.74** | 0.68 | **0.73** | 0.69 |

*Table 8.* Evaluation results for our CoT bypass described in Section 3.2, varying thresholds of normalized entropy obtained from final-answer probing at early CoT positions. The CoT bypass ratio for each task is reported. **Avg.** denotes the average bypass ratio, and **Perf.** indicates the average change in task performance measured by absolute accuracy. Result discussions are addressed near Table 4.

| | Parity | Cycle | Subsum | GSM8K | MATH | AIME | MuSR | Zebra | CSQA | MMLU | QuALITY | GPQA | **Avg.** | **Perf.** |
|---|---|---|---|---|---|---|---|---|---|---|---|---|---|---|
| | | | | | | In-Domain LLM | | | | | | | | |
| Th=0.02 | 0% | 0% | 0% | 0% | 0% | 0% | 1.2% | 3.6% | 16.6% | 11.2% | 26.2% | 2% | 5.07% | -0.15 |
| Th=0.05 | 0% | 0% | 0.2% | 0% | 0% | 0% | 2.8% | 11.2% | 27.4% | 22% | 41.4% | 4.4% | 9.12% | -0.32 |
| Th=0.1 | 0% | 0% | 0.2% | 0% | 0% | 0% | 8.2% | 18.4% | 40.2% | 30.4% | 55.6% | 7% | 13.33% | -0.47 |
| Th=0.2 | 0% | 0% | 0.2% | 0% | 0% | 0% | 27% | 34.6% | 65% | 45% | 75.8% | 12% | 21.63% | -1.42 |
| | | | | | | Off-the-Shelf LLM (Qwen3-32B) | | | | | | | | |
| Th=0.02 | 0% | 0% | 0% | 0% | 0% | 0% | 0% | 0% | 6.8% | 5% | 0.6% | 0.2% | 1.05% | -0.03 |
| Th=0.05 | 0% | 0% | 0% | 0% | 0% | 0% | 0.2% | 0% | 10.4% | 8.4% | 2% | 0.4% | 1.78% | -0.03 |
| Th=0.1 | 0% | 0% | 0% | 0% | 0% | 0% | 0.2% | 0% | 16.2% | 12.4% | 3.2% | 1.2% | 2.77% | -0.03 |
| Th=0.2 | 0% | 0% | 0% | 0% | 0% | 0% | 14.4% | 0% | 28.8% | 20.2% | 7.6% | 3.2% | 6.19% | -0.37 |

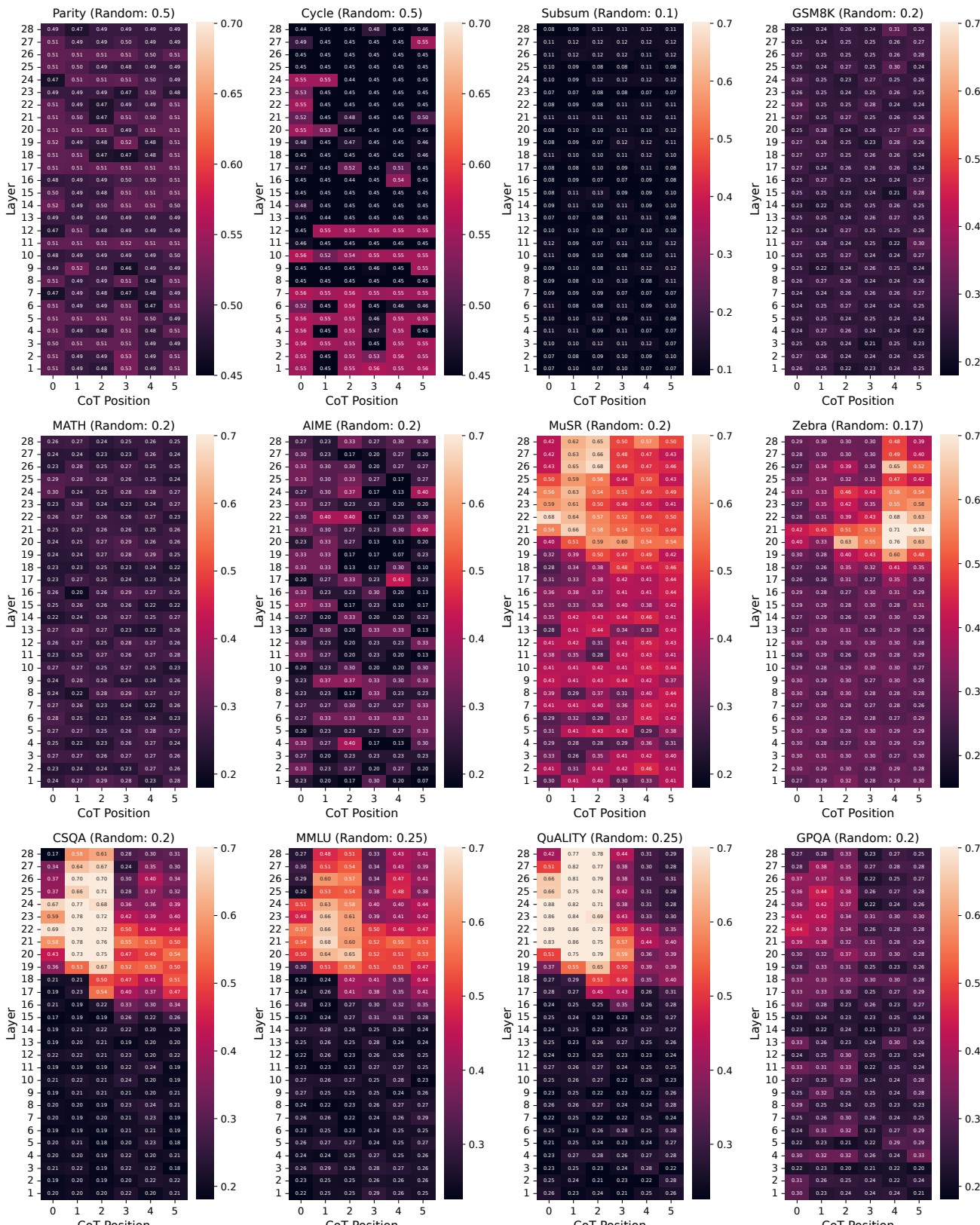

*Figure 13.* Probing for final answers: averaged accuracy with In-Domain LLM for the first six tokens along CoT trajectories, measured across Transformers layers and tasks. Result discussions are addressed in Section 2.5.1.

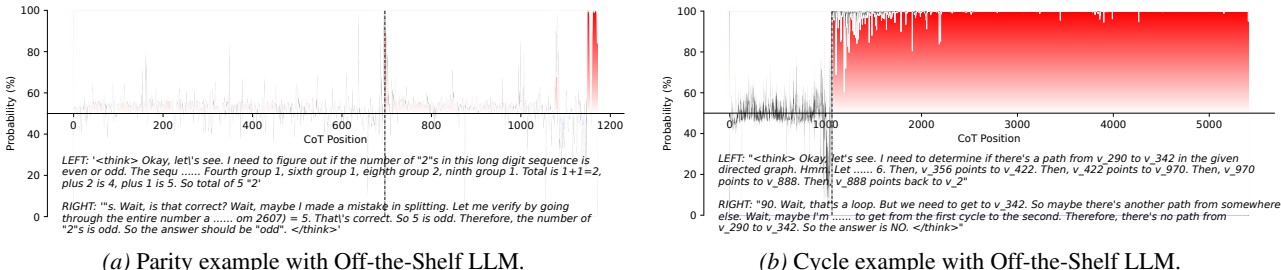

*Figure 14.* Probing for final answers: averaged accuracy with Off-the-Shelf LLM (Qwen3-32B) for the first six tokens along CoT trajectories, measured across selected Transformers layers and tasks.

*(a) Parity example with Off-the-Shelf LLM.*    *(b) Cycle example with Off-the-Shelf LLM.*

*Figure 15.* Examples of final-answer probing probabilities along CoT trajectories with Qwen3-32B (random guessing is at 50%). The vertical dashed line indicates the position at which accuracy first spikes. "*LEFT*" and "*RIGHT*" at the bottom illustrate the reasoning details right before and after the accuracy spike, respectively. Examples with In-Domain LLM are addressed in Figure 2.

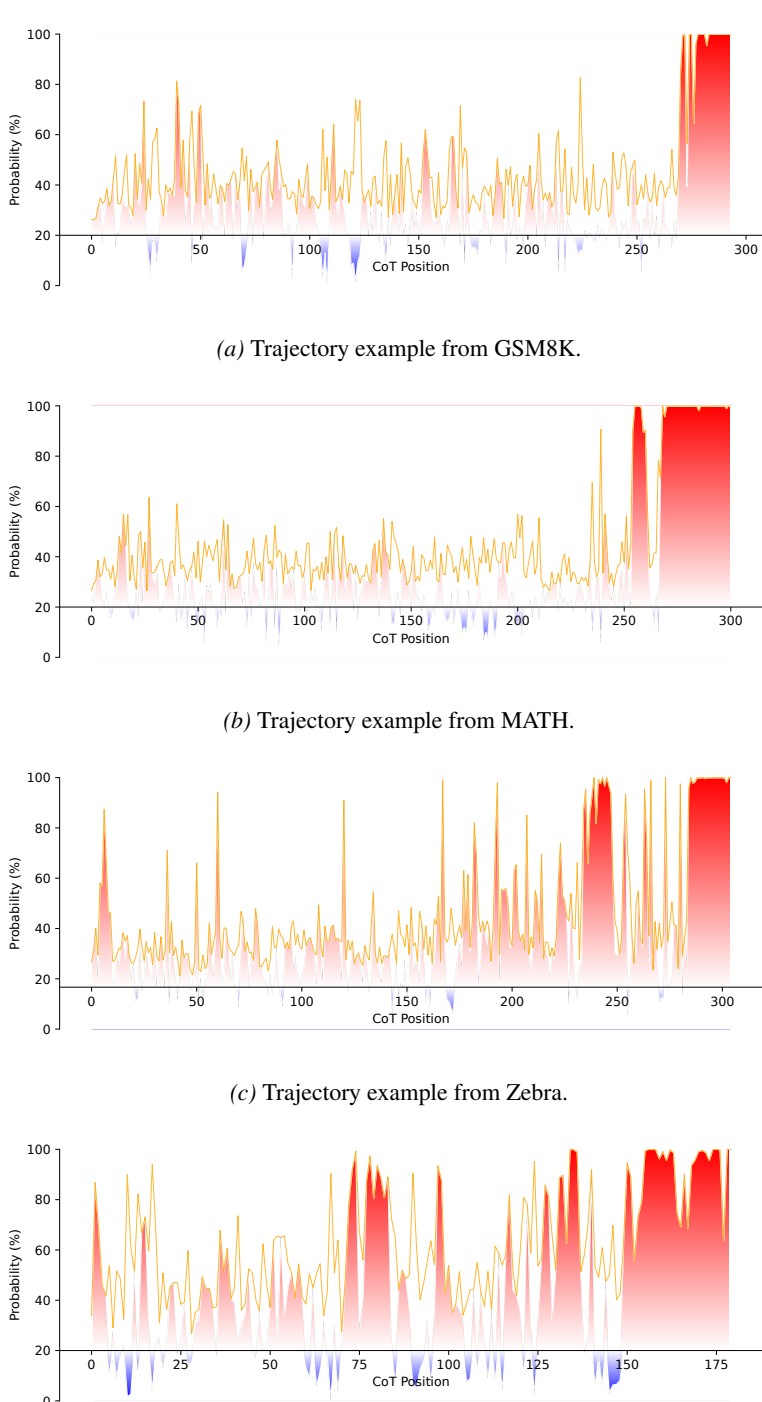

*(a)* Trajectory example from GSM8K.

*(b)* Trajectory example from MATH.

*(c)* Trajectory example from Zebra.

*(d)* Trajectory example from CSQA.

*Figure 16.* Examples of final-answer probing probabilities along CoT trajectories with In-Domain LLM. The yellow line denotes the maximum probability over the answer space at each step. For tasks beyond explicit compositional reasoning, accuracy spikes also occur sparsely. Especially for mathematical and logical reasoning, the final answer emerges towards the end of the reasoning, indicating a myopic planning horizon. More discussions are addressed near Figure 2.

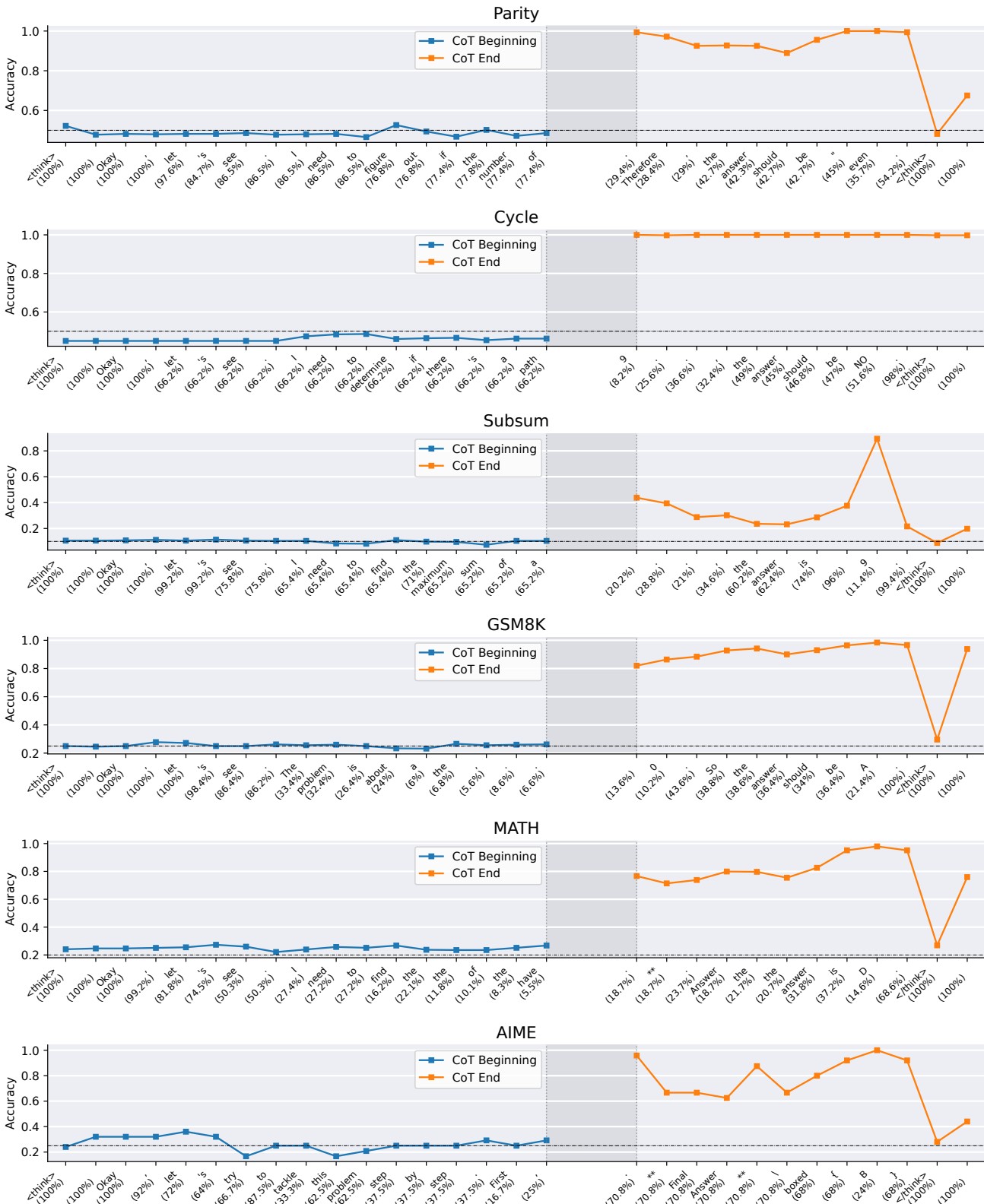

*Figure 17.* Probing for final answers: averaged accuracy with Qwen3-32B along CoT positions. The most frequent token at each position is annotated with its occurrence frequency (the remaining 6 tasks are shown in Figure 18).

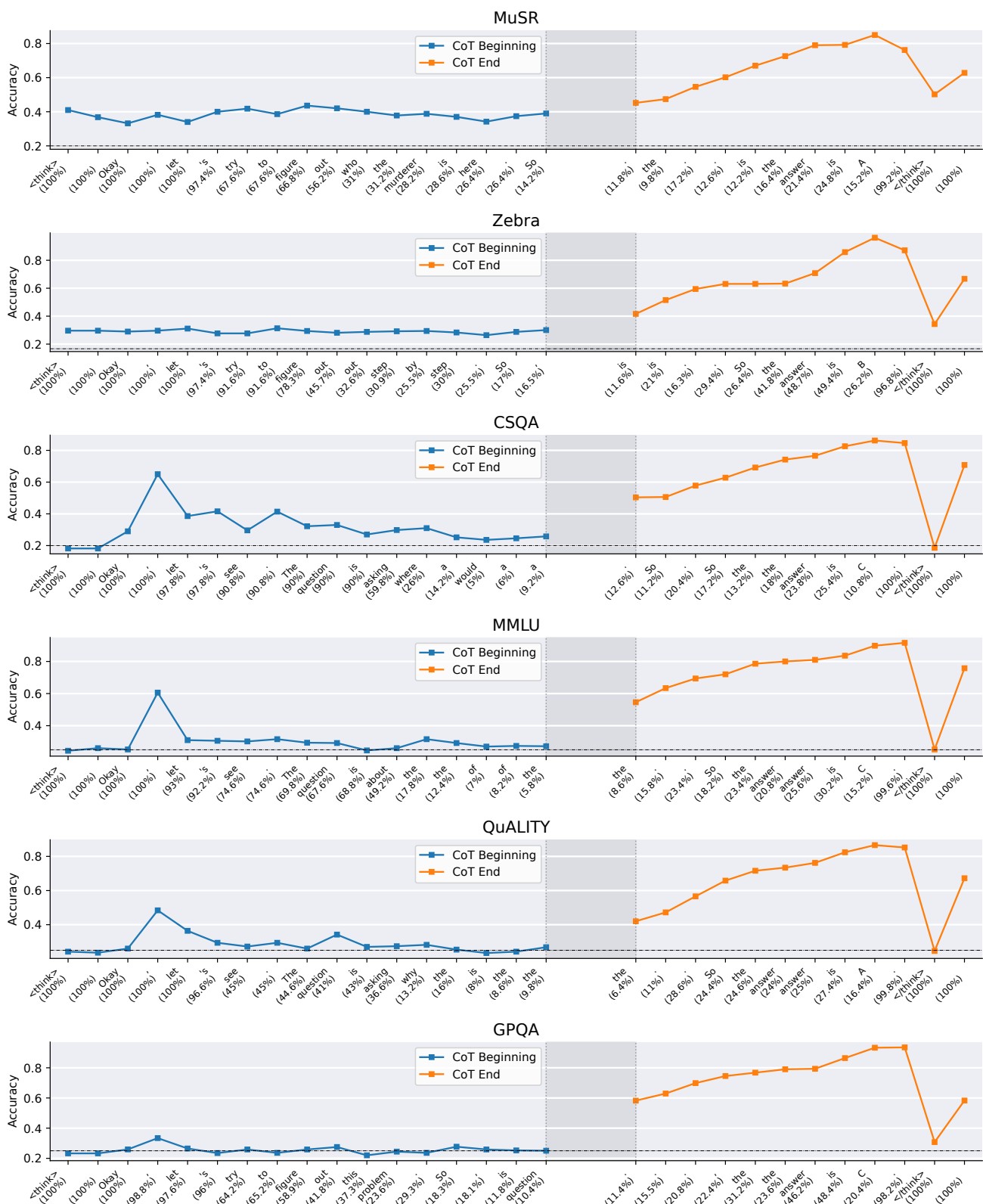

*Figure 18.* Probing for final answers: averaged accuracy by Qwen3-32B along CoT positions. The most frequent token at each position is annotated with its occurrence frequency. Result discussions are addressed near Figure 3.

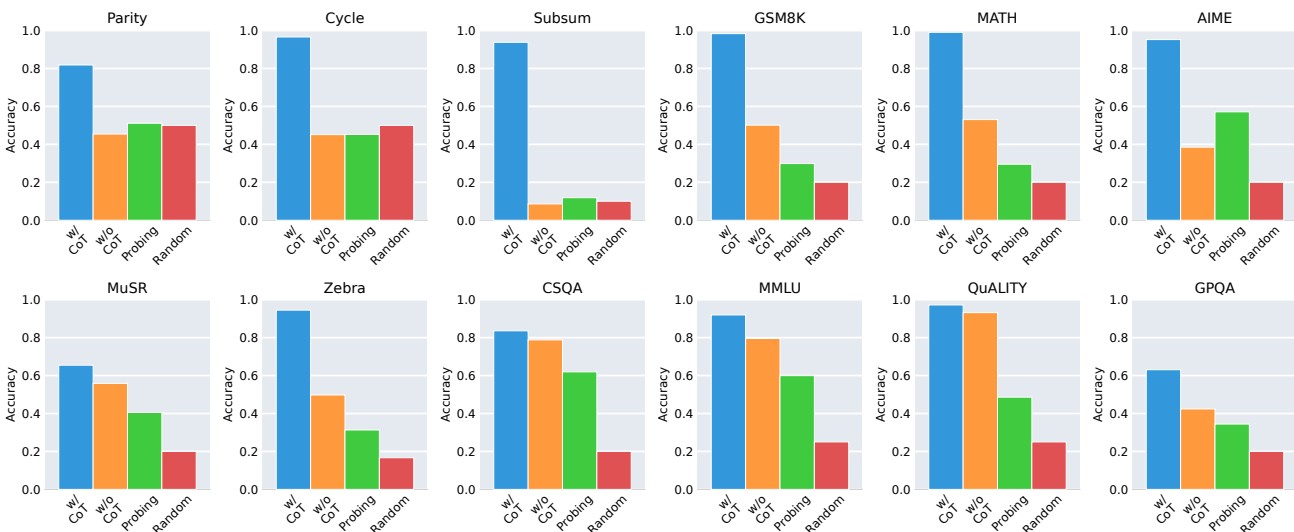

*Figure 19.* Task accuracy comparison for Off-the-Shelf LLM (Qwen3-32B) under four settings: using thinking mode (**w/ CoT**); using non-thinking mode (**w/o CoT**); the best probing accuracy among initial CoT positions (**Probing**); the random-guess baseline (**Random**).

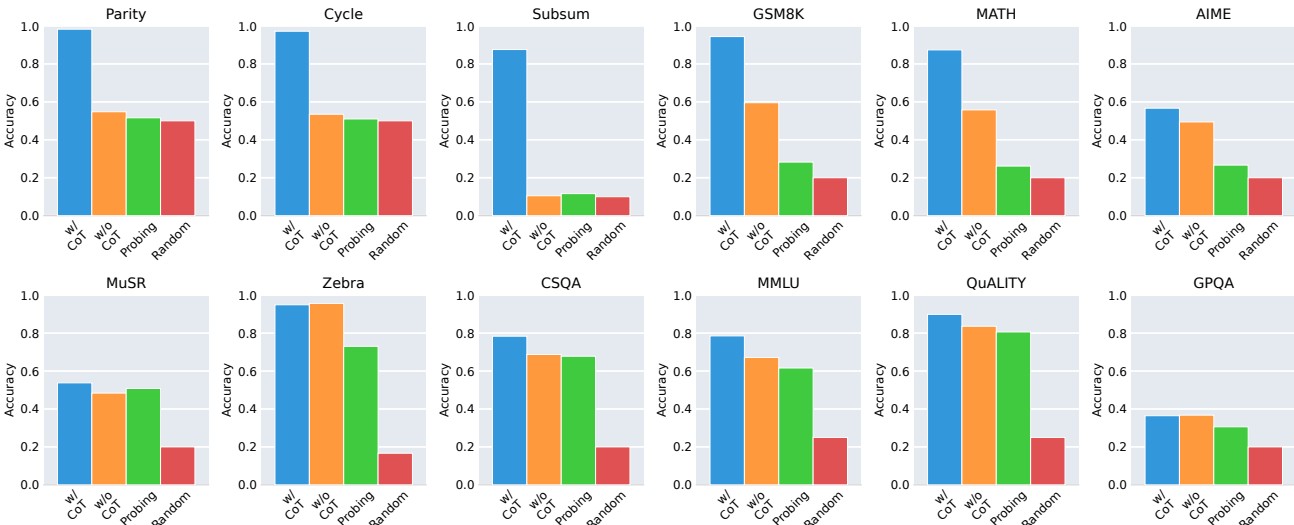

*Figure 20.* Task accuracy comparison for In-Domain LLM under four settings: standard inference with learned CoT (**w/ CoT**); direct prediction by a separately trained model via naive supervised finetuning, without CoT learned (**w/o CoT**); the best probing accuracy among initial CoT positions (**Probing**); the random-guess baseline (**Random**). Result discussions are addressed near Figure 4.

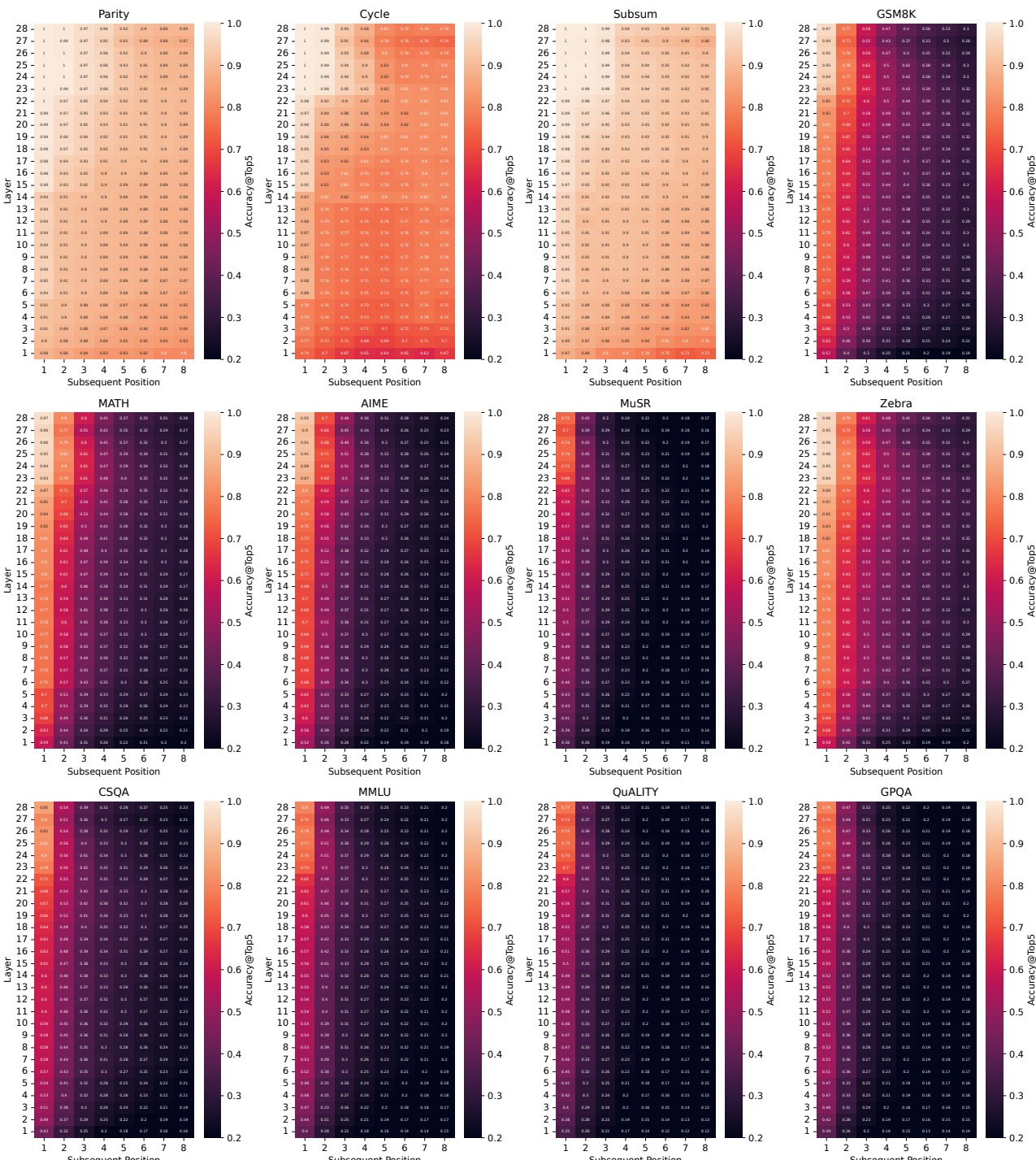

*Figure 21.* Averaged Top-5 Accuracy of subsequent token prediction with In-Domain LLM, across Transformers layers and subsequent positions (up to the 8th following position). Result discussions are addressed near Figure 5.

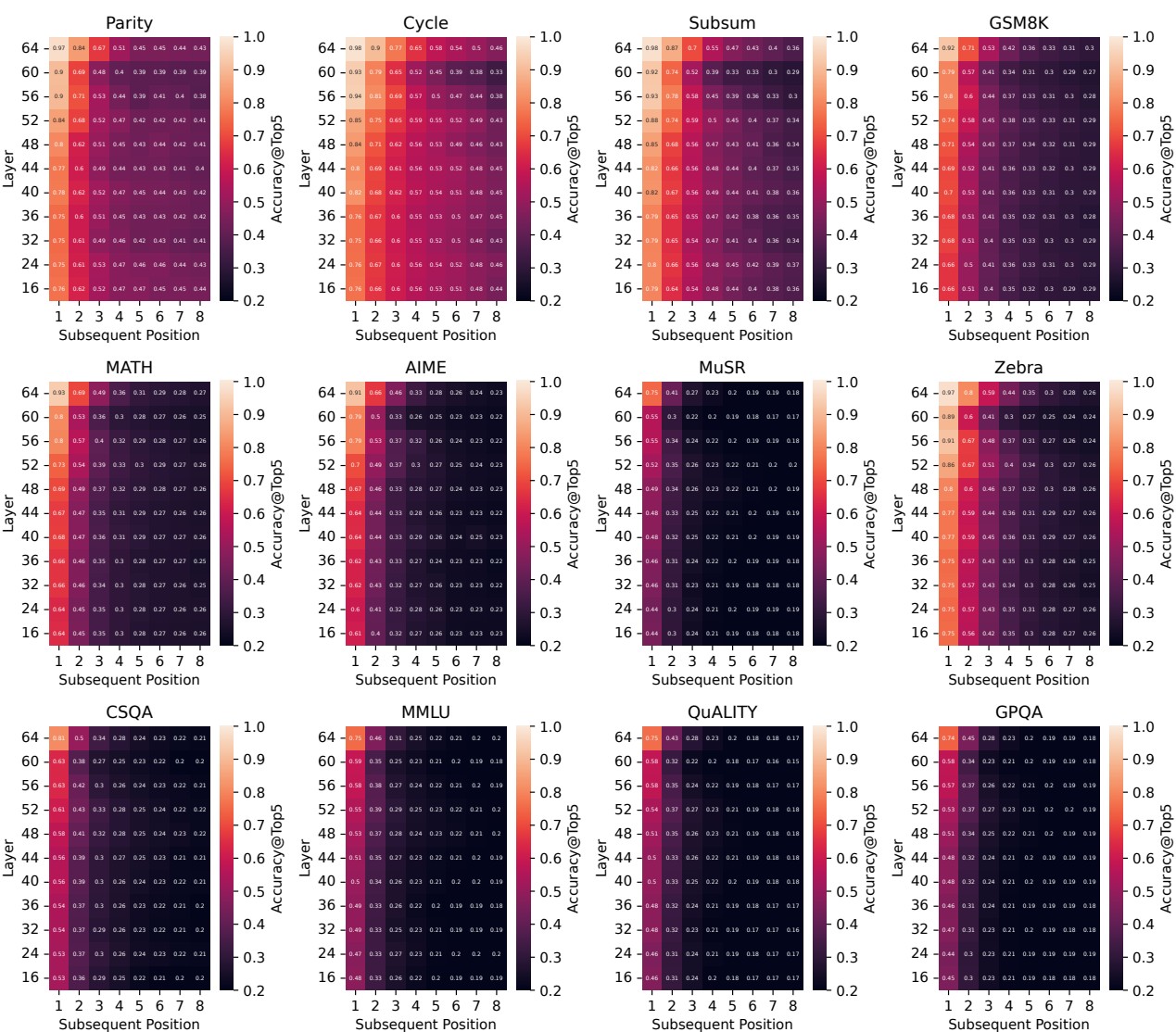

*Figure 22.* Averaged Top-5 accuracy for subsequent token prediction with Off-the-Shelf LLM, across selected Transformers layers and subsequent positions (up to the 8th following position).

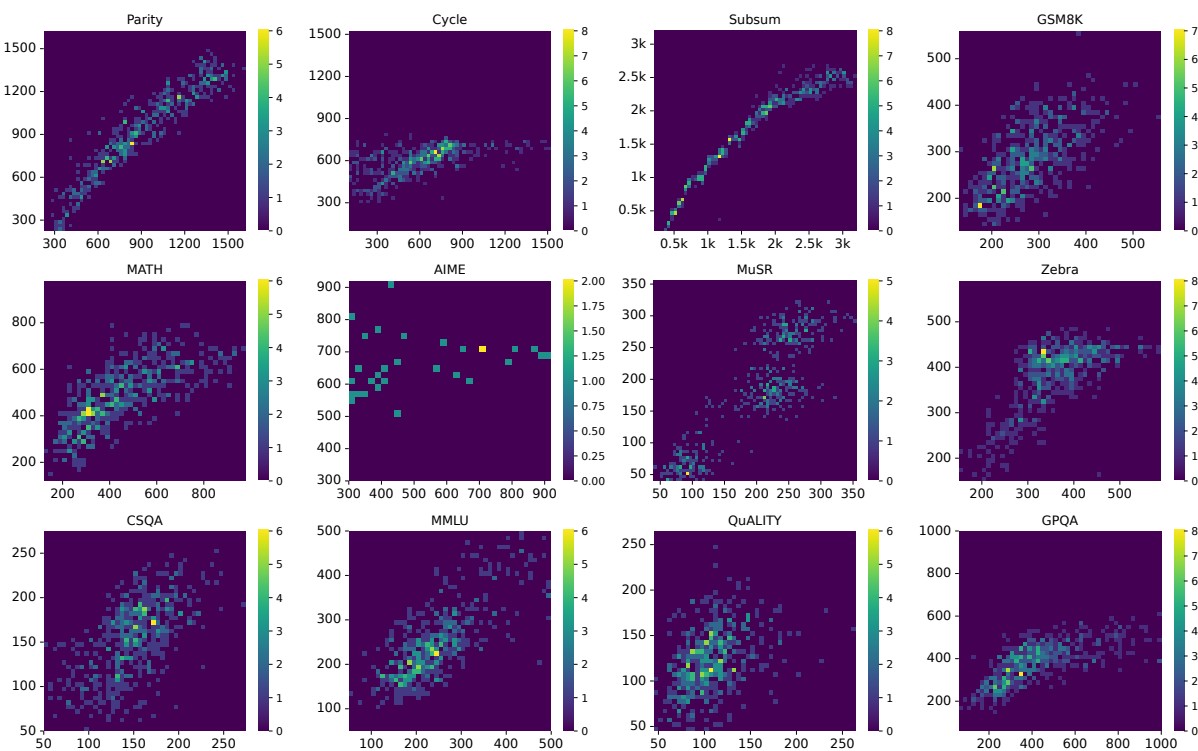

*Figure 23.* Probing for reasoning length: heatmap of the predicted length (*y*-axis) against the actual length (*x*-axis) for In-Domain LLM.

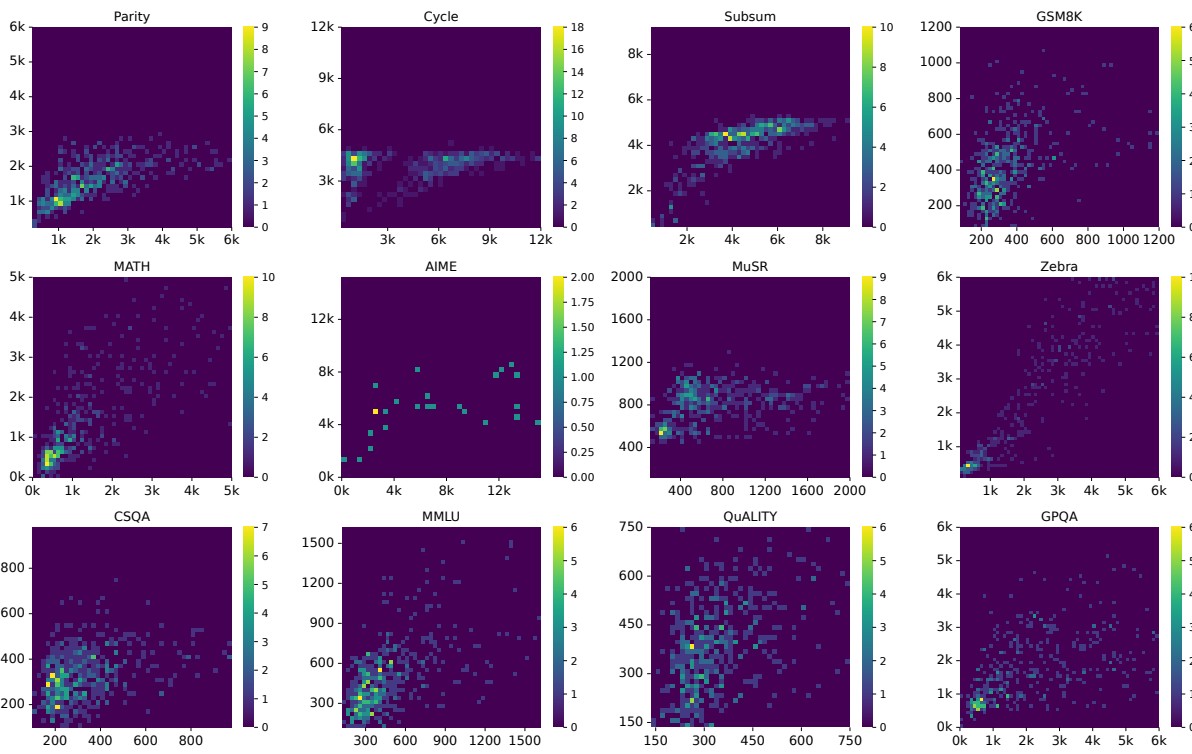

*Figure 24.* Probing for reasoning length: heatmap of the predicted length (*y*-axis) against the actual length (*x*-axis) for Off-the-Shelf LLM. Result discussions are addressed near Figure 6.

