# OpenReview forum: "How Far Ahead Do LLMs Plan? Uncovering the Latent Horizon in Chain-of-Thought Reasoning"
_ICML.cc/2026/Conference — ICML 2026 regular_

### Official Review · Reviewer_26j4 · 2026-03-01

**Soundness:** 3
**Presentation:** 3
**Significance:** 2
**Originality:** 2
**Overall Recommendation:** 4
**Confidence:** 4

**Summary:**

This paper attempts to deepen the understanding between LLM’s internal states and its verbalized reasoning trajectories. They propose a probing method Tele-Lens to analyze the hidden states across diverse task domains. They conclude that LLMs exhibit a myopic horizon without precise global planning. Finally, based on the observations found in this paper, they achieve the automatic recognition of CoT bypass.

**Compliance With Llm Reviewing Policy:**

Affirmed.

**Final Justification:**

I thank the authors for the rebuttal and for the clarifications. I increase my score. I hope the authors can well-incorporated these insights.

**Key Questions For Authors:**

Please refer to the weaknesses.

**Limitations:**

yes

**Strengths And Weaknesses:**

Strengths：
1. This paper studies an interesting problem of LLM’s internal states and its verbalized reasoning trajectories. The observations are well-presented.
2. This paper is well-written and easy to follow.

Weaknesses：
1. In Figure 1, the authors perform final-answer probing for the first five tokens. This setting may be over-strict for most LLMs as the models just begin to answer and may conduct planning. Thus, I am not sure if this setting is appropriate.
2. In Section 2.1, the authors study the planning ability of LLMs in predicting the Reasoning length and Final answer. These two metrics may be difficult for LLMs to predict, thus weakening the importance of this paper.
3. In Section 3.2, the CoT bypass method is straightforward and not new. The fixed probing class dimension C is also a hyper-parameter for different datasets.

---

> ### Author Rebuttal · Authors · 2026-03-30
>
> We sincerely thank the reviewer for the valuable feedback and thoughtful questions. We address the three comments in detail as follows.
>
> **Final-answer probing for the first five tokens may be overly strict for LLMs.**
>
> - We understand the concern that restricting the analysis to only the initial CoT positions may appear too stringent. In fact, **our probing evaluation does include all CoT tokens along the full reasoning trajectories**, as described in Appendix A.5 during dataset preparation ("*retain hidden states of all tokens along CoT trajectories within a maximum length of 16,384 for the test split*"). We apologize for the ambiguity in the current presentation and will make this point explicit in the final revision to avoid confusion.
> - More importantly, the current paper already presents final-answer probing results **over full CoT trajectories**. In the main paper, both **Figure 2 and Figure 3** present the final-answer probing dynamics along full trajectories. In the appendix, **Figure 14 and 15** visualize final-answer probing probabilities on more CoT trajectories across diffrent tasks; **Figure 16 and 17** depict the final-answer probing accuracy at more CoT positions along the trajectories.
> - We agree with the reviewer that probing only the first five CoT tokens would not be sufficient by itself, and we thank for the thoughtful comment. In our Section 2.5.1, our finding about the myopic planning horizon is derived **not only by the first five tokens**, but also through the **qualitative observations over full trajectories**, as well as **quantitative analysis in Table 1**.
>
> **The Reasoning length and Final answer may be difficult for LLMs to predict, weakening the importance of this paper.**
>
> - These two probing dimensions are indeed challenging for LLMs. However, we do respectfully believe that they can be important to reveal the relationship between explicit CoT and the underlying latent dynamics. In particular, they serve as strong indicators of internal planning: if LLMs maintained a precise global plan early in the reasoning process, then the strength of that plan should **in principle be measurable before explicit CoT fully unfolds**. Our empirical findings instead support a more local planning horizon rather than a strong global one.
> - Beyond this motivation, the reasoning length and final answer predictions are also standard targets in related probing studies, e.g. [1] and [2], where prior works states the final answers can be probed with high accuracy. Our work further **offers a deeper insight**: such high accuracy does not stem from a well-formed global plan, but rather from **local heuristics that are effective only on simpler tasks**.
>
> **The CoT bypass method is straightforward and not new, and a hyperparameter is involved.**
>
> - We agree that our CoT necessity estimation method is simple and straightforward. As noted in our responses to Reviewer 6mBx, the CoT bypass experiments mainly serve a demonstrative purpose on exploiting internal planning signals, which we term it as a *proof-of-concept* in Introduction and Section 3.2.
> - That said, to the best of our knowledge, this is the first CoT bypass method motivated by an explicit analysis of CoT synergy and latent planning signals, **grounded in a comprehensive study of CoT dynamics** across diverse tasks. In particular, Tables 4 and 8 show that latent signals in hidden states can reliably distinguish whether a CoT rollout is necessary, effectively reflecting the difficulty of the input.
> - We would also like to highlight that CoT bypass is not the main application we study. A central practical contribution of our exploitation analysis is **improved uncertainty calibration for CoT**. In Section 3.1, based on the proposed Wooden Barrel principle, we show that a small set of pivot tokens can better capture the uncertainty of full CoT trajectories (Table 2), leading to up to 6% improvement across all three general uncertainty metrics (Table 3).
>
> We sincerely hope the above clarifies the original concerns. Please let us know whether there are any remaining questions or concerns. We thank the reviewer again for the time and effort in reviewing this work, and we would be very happy to engage in further discussion during the rebuttal period.
>
> Thanks,
>
> Authors of the submission
>
> **References**
> - [1] Dong et al. (ICML'25) *Emergent Response Planning in LLMs*.
> - [2] Gottesman and Geva. (EMNLP'24) *Estimating Knowledge in Large Language Models Without Generating a Single Token*.

---

> > ### Author Rebuttal · Reviewer_26j4 · 2026-04-03
> >
> > Thanks for your response. I hope the authors can give more insights for my second question.

---

> > > ### Author Response · Authors · 2026-04-05
> > >
> > > We greatly appreciate the reviewer's feedback. We agree that clarifying the second question is important to illustrate our probing motivation, and we would like to discuss the significance of probing the reasoning length and final answer from **four aspects**.
> > >
> > > First, we would like to reiterate that the probing of these targets (final answer and reasoning length) is performed on **ALL CoT tokens**, not limited to the first five tokens, as we have clarified in our initial response. In our experiments, the question is **not whether they can be predicted**, but rather **when they can be predicted** along the CoT trajectory.
> > >
> > > Second, predicting these targets are not necessarily "difficult". Conversely, this work is **motivated by exactly by how "easy" they appear to be**, as reported by prior works. For instance, [1] reports that the reasoning length and final answer are probed with high accuracy on their designed task. [2], [3], and [4] find that LLM already "plans" their final answers prior to CoT emergence. Therefore in this work, we explicit examine the synergy between explicit CoT steps and its latent planning horizon, and our empirical results **reveal a deeper insight**, where the internal planning is actually *myopic*, and those final answer planning observed by prior works does not arise from precise planning (as described in Section 2.5.1).
> > >
> > > Third, the final answer prediction can be valuable because it serves as an anchor for characterizing the internal CoT dynamics. In CoT, **answer tokens can be viewed as a natural continuation of reasoning tokens**, rather than a separate object. By predicting final answers, one can assess **how early** and **how strongly** future outcome information is planned internally, which provides evidence about the strength and horizon of latent planning. In that sense, predicting final answers along CoT explicitly **captures the horizon of internal planning**, as depicted in Figure 2.
> > >
> > > Lastly, from a hypothesis-testing perspective, if LLM does have a precise global plan internally, the strength of such planning should **in principle be measurable** before CoT unfolds. Our empirical results suggest the lack of such long-term planning, which further **motivates our Section 3 that emphasizes the utilization such local latent signals**.
> > >
> > > We hope the above could further clarify the rationale on how these targets provide important insight into the central question of this work, and we will incorporate the relevant discussion into the final version of the paper.
> > >
> > > If there are any further aspects whose clarification would help support a stronger recommendation, we would be happy to address them promptly before the rebuttal period ends, as the reviewer's evaluation is extremely important to us. We thank the reviewer again for the thoughtful question, which we believe helps to present this work more clearly and precisely.
> > >
> > > Many thanks,
> > >
> > > Authors of the submission
> > >
> > > **References**
> > > - [1] Dong et al. (ICML'25) *Emergent Response Planning in LLMs*.
> > > - [2] Liu et al. (EMNLP'24) *Enhancing Language Model Factuality via Activation-Based Confidence Calibration and Guided Decoding.*
> > > - [3] Gottesman and Geva. (EMNLP'24) *Estimating Knowledge in Large Language Models Without Generating a Single Token*.
> > > - [4] Afzal et al. (ACL Finding'25) *Knowing Before Saying: LLM Representations Encode Information About Chain-of-Thought Success Before Completion*.

---

### Official Review · Reviewer_3azt · 2026-03-13

**Soundness:** 3
**Presentation:** 3
**Significance:** 3
**Originality:** 3
**Overall Recommendation:** 5
**Confidence:** 3

**Summary:**

This research focuses on the necessity of CoT with the approach to analyze the internal mechanism of LLM. The methods are Tele-Lens and probing. The preliminary experiment, the paper shows the probing result at the early phase of CoT for the final-answer and as a result of the experiment, paper states the hint for the existence of the global plan inside of LLM while it runs on CoT. The main experiments are two-fold: probing the entire phase of CoT and Tele-Lens to reveal the dynamism of CoT and visualize it on the figures. The paper also shows that potentially LLM can bypass CoT according to the probing result on the fly.

**Compliance With Llm Reviewing Policy:**

Affirmed.

**Final Justification:**

I appreciate the authors for the rebuttal and I increase my score as I mentioned in the last reply.

**Key Questions For Authors:**

1. Please justify/show the reason why the final-answer could be evidence to show the existence of the global plan.
2. If we assume the small language model—i.e., the model has less knowledge to solve—it is obvious that probing becomes hard to predict the final-answer at the early phase. Does the paper also conclude that the model has no global plan even though they take effort to search for solutions with CoT?

**Limitations:**

yes

**Strengths And Weaknesses:**

Strengths:
- This paper intensively uses the visualization rather than displaying just tables to reveal the dynamism of CoT.
- It is excellent that the paper shows the meta-viewpoint: LLM has (no/ a) global *plan* in the phase of CoT.
- It is solid to use Tele-Lens and a probing approach to investigate the LLM plan.
  - but , I could not agree with probing the model with the *final-answer*

Weakness:
- This paper states that the LLM has no global plan because of the difficulty of probing the model with the final-answer at the early phase in CoT. However, it should be distinct, from the LLM knowing the final-answer that the LLM has the global plan (plan is not the final answer/goal).
- The research question has already been stated in the prior study, and the questions stated in the paper come from the complementary perspective.

---

> ### Author Rebuttal · Authors · 2026-03-31
>
> We sincerely appreciate the reviewer for the valuable feedback, especially the raised question on "global plans". We believe the clarification below can further elucidate the positioning of this work.
>
> **Q1. Whether LLMs have global plans to solve problems.**
>
> We identify the key issue here is that the term "*planning*" is being used in **two different senses**.
>   - The reviewer’s interpretation: closer to **explicit or verbal planning**, i.e., whether the model can produce a reasoning process or steps of a solving plan in explicit language.
>   - The *planning* studied in this work: **internal planning in the latent dynamics of CoT**, i.e., whether, and over what horizon, the subsequent generation is internally organized during the autoregressive reasoning trajectory.
>
> As we described in Introduction, this work studies the synergy between explicit CoT steps and the underlying latent planning horizon. Accordingly, our obsevation of "No Global Plan" **does NOT mean LLMs cannot form explicit plans** to solve problems. Rather, our empirical evidence suggests that the **CoT is formed with local-horizon latent planning, rather than a strong global latent plan fixed early in the trajectory**.
>
> Given this distinction, we agree with the reviewer that LLMs are capable of producing planning steps to solve complex problems. However, that capability concerns a different question from the one examined in this work: being able to express an explicit plan does not necessarily imply a strong latent planning signal internally. Our work is aimed precisely at studying this internal aspect.
>
> We do appreciate the reviewer’s comment because it helps sharpen this distinction, and we will polish our Introduction to make it more explicit. We would be very glad to discuss any follow-up questions or suggestions during the rebuttal period.
>
> **Q2. Whether final answer probing reflects the existance of global plans.**
>
> Under our view of internal planning, the final answer probing is useful because it serves as a valuable anchor for probing the internal CoT rollout. In CoT, **answer tokens are a natural continuation of reasoning tokens rather than a separate object**. Therefore, probing final-answer information is one way to **evaluate how early and how strongly future outcome information is represented internally**, which in turn provides evidence about the strength and horizon of latent planning.
>
> This also clarifies the question of whether a model may still search for solutions with CoT even if early probing does not strongly recover the final answer. We believe the answer is yes. **A model can still carry out effective reasoning and reach the correct solution, even when its internal planning horizon is relatively local**. For example, for the Parity task shown in Figure 2, the model may update intermediate state step by step and eventually reach the correct answer, while hidden states mainly encode local transitions rather than a strong long-range commitment to the final outcome. In this case, the final-answer probing suggests a more local latent planning horizon, rather than the absence of reasoning ability.
>
> **Q3. The research question has already been stated in the prior study.**
>
> Probing LLM hidden states is indeed not new. As our Introduction mentions, several prior works have shown that early hidden states can encode information about subsequent generation, in some cases even enough for the final answer to be decoded directly (L23). However, this work substantially broadens the scope on both the **probing** side (Section 2), but also on the **exploitation** of such latent signals (Section 3):
>
> - *Probing*: we propose our probing model, Tele-Lens, along with three types of probing tasks across 12 datasets in total. Our results provide a more unified view of previously complementary observations and suggest a deeper insight: **prior high probing accuracy does not stem from a well-formed global plan, but rather from local heuristics that are effective only on simpler tasks**.
>
> - *Exploitation*: we further exploit latent planning signals to estimate two CoT-related properties. Section 3.1 utilizes these signals to better estimate the **uncertainty of CoT trajectories**, and Section 3.2 estimates when CoT is unnecessary for deriving the final answer, enabling **automatic CoT bypass** without performance degradation.
>
> To our knowledge, we are not aware of prior work that highlight both in-depth analysis of LLM probing and a systematic utilization of latent signals in this way. We respectfully believe that this work can bring a unique contribution to the research community.
>
> We hope the above could clarify the original concerns. We thank again for the thoughtful questions, which we believe are important to discuss in the final paper towards a more precise presentation. Please let us know if there are any remaining concerns, and we would be sincerely happy for any discussion during the rebuttal period.
>
> Thanks,
>
> Authors of the submission

---

> > ### Author Rebuttal · Reviewer_3azt · 2026-04-06
> >
> > Q1. Whether LLMs have global plans to solve problems.
> >
> > Thank you for clearing up the misunderstanding about this. I understood it from the point of view of whether it is on latent dynamics in a more theoretical sense, rather than explicit planning in the engineering and real-life application layer. This concern has been resolved.
> >
> > Q2. Whether final answer probing reflects the existence of global plans.
> >
> > This is also not an explicit planning in the engineering and real application layer, but rather from the perspective of seeing whether it is on latent dynamics in the sense of theory, and I understood that we would see how the continuation from the state of the observation to the final token is.
> >
> > Q3. The questions in the study have already been raised in previous studies.
> >
> > The superficial method acknowledged the similarity with the existing probing, but the rebuttal that the conditions and methods for probing the potential state were fully convinced.

---

> > > ### Author Response · Authors · 2026-04-07
> > >
> > > We sincerely thank the reviewer for the very encouraging follow-up, and for confirming that the three concerns have been addressed! The original questions and comments regarding the "global plans" have been very important in helping us clarify the paper, sharpen our claims, and improve the overall presentation.
> > >
> > > And we greatly appreciate the careful reading of our rebuttal. If there are any further questions or aspects that would support a stronger recommendation, we would be very happy to address them promptly before the rebuttal period ends. If the reviewer feels that all concerns have been resolved, we would sincerely appreciate an updated recommendation to reflect this current assessment, as the reviewer’s evaluation is extremely valuable to us.
> > >
> > > We sincerely thank the reviewer again for the time and effort in reviewing this work.
> > >
> > > Thanks,
> > >
> > > Authors of the submission

---

### Official Review · Reviewer_FqzY · 2026-03-16

**Soundness:** 2
**Presentation:** 1
**Significance:** 2
**Originality:** 2
**Overall Recommendation:** 4
**Confidence:** 2

**Summary:**

I think the paper studies whether large language models actually plan globally during reasoning or instead rely on more local,  step by step  heuristics.


 The author then analyzes the reasoning trajectories and show evidence that model behavior is largely myopic rather than driven by a coherent global plan. The paper supports this claim through empirical analysis across reasoning tasks and model behacviories.

**Compliance With Llm Reviewing Policy:**

Affirmed.

**Key Questions For Authors:**

How sensitive are the results to prompting strategies or reasoning formats (e.g., different CoT prompts)?
What happen if I have latent LLMs?

**Strengths And Weaknesses:**

paper raises an interesting and important question about the nature of reasoning in LLMs, and the empirical analysis provides useful observations about how reasoning unfolds in practice. The framing is clear and the experiments provide some evidence supporting the claim that models behave in a largely local manner.

The major downside is the writing, and the presentation, whcih I believe can be polished. Also, the practical implications of the findings could be articulated more clearly.

Also, so more illustrative examples or visulaization will help  the readers to understand the paper better.

---

> ### Author Rebuttal · Authors · 2026-03-30
>
> We sincerely thank the reviewer for the valuable feedback and for recognizing the significance of our proposed research on unfolding CoT's latent planning dynamics. We address the three comments in detail as follows.
>
> **The writing can be polished, with more illustrative examples or visualization**.
>
> - We apologize for any confusion caused by unclear descriptions; if the reviewer brings up any specific cases in mind, we will make sure to clarify them further during the rebuttal period, and revise the final version accordingly.
> - In the main page, we tried to include intuitive **plotting** for easier understanding, as recognized by Reviewer 3azt that **we intensively use visualization**, instead of just stacking tables. We also prepared a **full example for each task in Appendix A.1** (12 examples in total), along with dataset preparation, statistics and hyperparameters in Appendix A.2-A.6. Appendix B provides details of our LLM backbones, with **CoT examples in Appendix B.3**.
> - Due to the page limit, the 8-page main text cannot fit in all details. Our current appendix contains **19 pages with 12 full figures**. When the additional page is available, we plan to move certain details back to the main pages, especially empirical details from Appendices A.3 and D, to make the experimental setting easier to follow.
>
> **The practical implications of the findings could be articulated more clearly**.
>
> - We agree that the significance of exploiting latent signals of CoT dynamics can be further expanded. Our prior discussions were mainly placed in Appendix E. We will move this part back to the main page, with expanded discussions including:
>   - Utilizing latent signals for CoT compression, i.e. [1]
>   - Leveraging signals to steer CoT behavior, such as length control as in [2]
>   - Performing early stop of CoT, as in [3]
>   - Improving model training by identifying critical CoT tokens, as in [4]
>
> **How sensitive are the results to prompting strategies, and what happens for latent LLMs?**
>
> - In our preliminary experiments, we found that different prompt variants using Qwen3 LLMs lead to very similar probing results. More importantly, in our final design, we explicitly include an **In-Domain LLM setting** in addition to Off-the-Shelf LLMs such as Qwen3-32B, in order to **reduce potential confounding factors from prompting strategies and model families**. As described in Section 2.3, the In-Domain LLM undergoes **task-aware RL training via GRPO**, thus serving as an "upper bound" for the internal planning, producing much more decisive reasoning paths as discussed in L173 and Table 5. The task-aware RL training essentially removes the effect of prompt variation, and the empirical findings remain **consistent**, as shown by the full results in Figure 12-23.
> - For LLMs with latent thinking process, most related approaches are trained in a **distillation/compression manner** as in [5] and [6]. In that sense, their latent dynamics may still be expected to remain broadly consistent with those of regular CoT-based LLMs, since they are learned by compressing standard CoT trajectories. We have conducted additional inference using the open-source latent thinking model Coconut [5]; however, it did not produce sufficiently reliable outputs for probing analysis, as it failed to follow instructions on most of our tasks. We are actively exploring if there are any open-source works that are suitable for our probing.
>
> We hope the above could address the original concerns, and please let us know any remaining issues that can be addressed for an increased evaluation score. We sincerely thank the reviewer again for the time and efforts in reviewing this work.
>
> Thanks,
>
> Authors of the submission
>
> **References**
> - [1] Li et al. (ICLR'26) *Compressing Chain-of-Thought in LLMs via Step Entropy*.
> - [2] Sheng et al. (NeurIPS'25) *On Reasoning Strength Planning in Large Reasoning Models*.
> - [3] Afzal et al. (ACL Finding'25) *Knowing Before Saying: LLM Representations Encode Information About Chain-of-Thought Success Before Completion*.
> - [4] Huang et al. (ACL'25) *Enhancing Chain-of-Thought Reasoning with Critical Representation Fine-tuning*.
> - [5] Hao et al. (COLM'25) *Training Large Language Models to Reason in a Continuous Latent Space*.
> - [6] Shen et al. (EMNLP'25) *CODI: Compressing Chain-of-Thought into Continuous Space via Self-Distillation*.

---

> > ### Author Rebuttal · Reviewer_FqzY · 2026-04-05
> >
> > thx

---

> > > ### Author Response · Authors · 2026-04-05
> > >
> > > We sincerely thank the reviewer for the positive follow-up and for confirming that the original concerns have been addressed. We remain grateful for the thoughtful questions and suggestions that help to strengthen the presentation of this paper. If there are any further questions whose clarification would support a stronger recommendation, we would be very happy to address them promptly before the rebuttal period ends.
> > >
> > > If the reviewer feels that all concerns have been resolved by the clarification, we would also greatly appreciate an updated recommendation to reflect the current assessment, as the reviewer’s evaluation is extremely valuable to us. We again sincerely thank the reviewer for the time and efforts in reading our responses.
> > >
> > > Many thanks,
> > >
> > > Authors of the submission

---

### Official Review · Reviewer_6mBx · 2026-03-18

**Soundness:** 3
**Presentation:** 3
**Significance:** 2
**Originality:** 3
**Overall Recommendation:** 4
**Confidence:** 4

**Summary:**

This paper studies whether the CoT of thinking LLMs is conducted with a global plan or in the step-by-step manner. The authors propose Tele-Lens as a method to predict the future reasoning tokens and final answers, utilizing hidden states of the reasoning tokens. The paper finds thinking LLMs adopt myopic planning in their thinking process, which means that thinking LLMs do not have the global plan or the final answer at the early stage of their CoT, instead LLMs reasoning only predict next several token information.

**Compliance With Llm Reviewing Policy:**

Affirmed.

**Final Justification:**

The author's rebuttal partially solved my concerns. My main concern is still that this observation is intuitive and not novel. Also, its CoT necessity application is then limited. However, given their comprehensive experiments, I'd like to give a weak accept.

**Key Questions For Authors:**

Please see weaknesses

**Limitations:**

No limitation is discussed. Suggestions for improvement:
Conducting necessary experiments on the adapter design and training ablations to further make sure the observation and conclusion is not overclaimed. Discuss more on the potential applications of the observed insights and more thorough experiment verification on necessity estimation.

**Strengths And Weaknesses:**

[S1] The problem studied is interesting and important. Studying thinking LLMs from the perspective of probing gives a better understanding and insights of the CoT process.

[S2] The presentation is good. The paper is well written and the idea is clear to follow.

[S3] The empirical verification of the idea is comprehensive. The experiments include diverse benchmarks and 12 tasks in total, making the proposed observation and insights more sound. Nevertheless, it will be greater to see further experiments on agent tasks.


[W1] My main concern about the paper is that the findings of myopic planning may be overclaimed. The observed probing failure can also be due to: 1) the low-rank adapter itself, in the terms of its architecture, may not be able to capture the global planning information in higher latent space; 2) even the adapter architecture is good enough, will the 5000 steps of training and early stopping be enough for training an very accurate probing model?

[W2] Another concern is that although the observations are interesting, their application is very limited. The authors propose two applications derived from their methods: calibration and CoT necessity estimation. For necessity estimation, since this is a well studied area and there are many recent works on adaptive thinking/efficient thinking of LLMs, comparison with some baselines should be included. Also, as is stated by the author, current necessity estimation aims to probe the final answer but may be hard to generalize to more open-ended questions.

---

> ### Author Rebuttal · Authors · 2026-03-31
>
> We sincerely appreciate the reviewer’s valuable feedback and suggestions for improvement, which we believe could help strengthen the empirical observations made in this paper. We address the two comments in detail below.
>
> **Make sure the findings of myopic planning is not overclaimed.**
>
> - We understand the concern that the low-rank adapter itself may not be sufficiently expressive to uncover the latent planning information encoded in hidden states, and we agree that additional empirical evidence is helpful for ruling out this factor. Accordingly, we conducted additional experiments under two enhanced settings as suggested:
>   - **Scaling adapter size**: as the original adapter is adopted low-rank as **256**, we further run experiments with adapters of rank **512, 1024, and 2048** respectively.
>   - **Scaling training budget**: as the original training is conducted for about **5K** steps with early stopping on the dev set, we now set the max training steps as **30K**, and force the early stopping patience as 10 dev set evaluation rounds, so that the early stop only happens when the dev set performance does not get improved for 10 consecutive rounds.
> - Experiments are conducted with both In-Domain LLM (7B) and Off-the-Shelf Qwen3 (32B), using hidden states of first five CoT positions from the last Transformers layer. The average accuracy is reported in the table below (we only show partial table here due to space limit).
>
> | Model | Setting | Parity | Cycle | Subsum | GSM8K | MATH |
> |---|---|---:|---:|---:|---:|---|
> | In-Domain LLM (7B) | R=256 (5K) | 0.49 | 0.46 | 0.10 | 0.26 | 0.26 |
> | In-Domain LLM (7B) | R=2048 (30K) | 0.49 | 0.48 | 0.11 | 0.26 | 0.25 |
> | Qwen3 (32B) | R=256 (5K) | 0.51 | 0.51 | 0.10 | 0.25 | 0.24 |
> | Qwen3 (32B) | R=2048 (30K) | 0.49 | 0.47 | 0.11 | 0.27 | 0.26 |
>
> - By scaling up both the adapter size and training budget, the probing performance remains **highly consistent** with those under the original setting. Especially, the performance stays *near random* for the three Explicit Compositional Tasks (Parity, Cycle, Subsum), and the accuracy does not differ much in other tasks, suggesting that the myopic planning observation is robust varying probing configurations.
>
> **The application on CoT necessity estimation is limited.**
>
> - Our experiments on CoT necessity estimation indeed mainly serves a demonstrative purpose on exploiting internal planning signals, achieving automatic CoT bypass on difficult inputs, which we term it as a *proof-of-concept* in Introduction and Section 3.2. Our aim of necessity estimation is to **distinguish whether a CoT rollout is necessary** to derive the final answer, which is quite different from "Compressing CoT" that aims to reduce thinking length: the former **examines the difficulty of the input space**, whereas the latter focuses on **increasing information density during reasoning**. Thus, they serve different purposes and are not quite directly comparable.
> - At the same time, we do agree that adding such a comparison can provide more insights. We also follow the research of efficient thinking closely (especially the direction of *latent CoT*), and we conduct an additional experiment using the *Soft Thinking* technique [2] with In-Domain LLM, reported as below. Compared with [2], our method shows stronger polarity with respect to input difficulty: it reduces more CoT on easier tasks, whereas [2] tends to produce a more uniform compression pattern.
>
> | Method | Parity | GSM8K | Zebra | CSQA | MMLU |
> |---|---:|---:|---:|---:|---:|
> | Ours | 0% | 0% | **18.4%** | **40.2%** | **30.4%** |
> | *Soft Thinking* | **1.9%** | **6.1%** | 5.3% | 13.3% | 12.8% |
>
> - It is also worth noting that [2] requires significantly more computation (**computing entropy of each CoT step**), while our method incurs negligible overhead (**computing entropy only once before CoT**), presenting a unique advantange of our necessity estimation direction.
>
> We would also like to highlight **CoT uncertainty estimation/calibration** in Section 3.1 as our main efforts exploiting internal planning signals. Our proposed strategy is directly applicable to general uncertainty metrics, and yields up to 6% absolute improvement, which we believe is quite significant, i.e. the gain is both larger and more robust than the recently proposed uncertainty metric [3].
>
> We hope the above could address the original concerns. Please let us know if there are any remaining questions or issues, and we would be happy to continue the discussion during the rebuttal period. We once again express our gratitude to the time and efforts in reviewing this work, which we believe will make a better version of the paper.
>
> **References**
> - [1] Dong et al. (ICML'25) *Emergent Response Planning in LLMs*.
> - [2] Zhang et al. (NeurIPS'25) *Soft Thinking: Unlocking the Reasoning Potential of LLMs in Continuous Concept Space*.
> - [3] Kang et al. (NeurIPS'25) *Scalable Best-of-N Selection for Large Language Models via Self-Certainty*.

---

> > ### Author Rebuttal · Reviewer_6mBx · 2026-04-02
> >
> > Thank you for the response. I still maintain positive about the work and decide to maintain my rating at this stage. Since you mentioned "distinguishing whether the cot is necessary", I feel adding experiments comparison to the final version of adaptive thinking methods [1,2] may be helpful.
> >
> > [1] Dynamic Early Exit in Reasoning Models (ICLR26)
> >
> > [2] Think or Not? Exploring Thinking Efficiency in Large Reasoning Models via an Information-Theoretic Lens (NeurIPS25)

---

> > > ### Author Response · Authors · 2026-04-05
> > >
> > > We greatly appreciate the reviewer's further feedback. The two suggested papers are relevant to our empirical demonstration of CoT necessity estimation, specifically:
> > > - Our CoT necessity estimation and these two papers **share the same broad objective, but with different design focus**. At a broad level, all three methods aim to identify when direct answering is sufficient, so as to avoid unnecessary CoT rollout. However, the two papers primarily study **early exit during CoT**, whereas our method focuses on **estimating on the input space directly**, so to determine whether direct answering is sufficient without invoking CoT.
> > > - As a result, there is a **substantial difference on computational cost**. Our method **computes entropy only once** prior to CoT emergence, whereas the two papers **computes entropy in proportional to CoT length**. Especially, [1] also incurs more overhead by inducing answers proactively during the reasoning process.
> > >
> > > Additionally, to provide an insightful illustration of the design difference, we further conduct additional experiments, comparing: (i) our CoT necessity estimation; (ii) CoT early exit by [1]; and (iii) CoT compression via *soft thinking* presented in our initial response. The results (compression rate) using our trained In-Domain LLM (7B) are shown in the table below.
> > >
> > > | Method | Parity | GSM8K | Zebra | CSQA | MMLU |
> > > |---|---:|---:|---:|---:|---:|
> > > | CoT Necessity by ours | 0% | 0% | **18.4%** | **40.2%** | **30.4%** |
> > > | CoT Early Exit by [1] | 0% | 0% | 0% | 0% | 0% |
> > > | CoT Compression by *Soft Thinking* | **1.9%** | **6.1%** | 5.3% | 13.3% | 12.8% |
> > >
> > > Notably, when CoT trajectories are already efficient, as produced by the In-Domain LLM, the effect of [1] becomes diminished, as its **early exit strategy relies on redundant verification behaviors** within CoT, triggerd by words such as "*wait*", while our trained In-Domain LLM rarely performs such additional checking.
> > >
> > > For reference, using Qwen3-32B, [1] can achieve higher compression rate when its CoT tends to repeatly perform self-check during reasoning, especially on more logic-oriented tasks, shown by the table below. Overall, the three lines of works reflect rather **different design rationales for improving LLM reasoning efficiency**.
> > >
> > > | Method | Parity | GSM8K | Zebra | CSQA | MMLU |
> > > |---|---:|---:|---:|---:|---:|
> > > | CoT Necessity by ours | 0% | 0% | 0% | **28.8%** | **20.2%** |
> > > | CoT Early Exit by [1] | **23.7%** | **39.4%** | **26.3%** | 20.7% | 18.8% |
> > >
> > > In the final paper, we will incorporate the above discussions into the section of CoT necessity estimation, which we believe helps to strengthen the corresponding scope. In the end, our scope of necessity estimation is **grounded by a systematic study of CoT's internal dynamics**, and is intended to motivate the exploitation of such latent CoT signals, but not particularly targeting state-of-the-art methods on LLM efficiency.
> > >
> > > We hope the above could clarify the rationale and the regarding comparison behind our CoT necessity estimation. If there are any further aspects whose clarification would help support a stronger recommendation, we would be happy to address them promptly before the rebuttal period ends.
> > >
> > > We sincerely thank the reviewer again for the time and efforts in reading our responses!
> > >
> > > Authors of the submission

---

### Decision · Program_Chairs · 2026-04-30

**Decision:**

Accept (regular)

**Comment:**

The paper analyzes the latent representation of LLM models when using Chain-of-Thoughts with the motivation of understanding their planning capabilities.

The discussion was rich and insightful. The reviewers were already interested in the empirical results. The discussion shed light on their interpretation.

I think this work deserves to be discussed at ICML.